# OpenReview forum: "LUSB: Formalizing and Benchmarking Unlearning Attacks and Defenses against Large Language Models"
_ICLR.cc/2026/Conference — Submitted to ICLR 2026_

### Official Review · Reviewer_ddTi · 2025-10-20

**Soundness:** 1
**Presentation:** 1
**Contribution:** 1
**Rating:** 0
**Confidence:** 3

**Summary:**

This work introduces Language Unlearning Security Benchmark (LUSB), a framework to formalize, evaluate and benchmark unlearning attacks and defenses on LLMs. Authors evaluate multiple combinations of attacks, defenses, unlearning methods, datasets and LLMs, showing that unlearning attacks enhance the vulnerability of LLMs to adversarial (backdoor, prompt injection or jailbreaking) attacks. Authors emphasize that existing defenses are not effective for unlearning attacks.

**Strengths:**

Authors evaluate multiple LLMs, datasets, unlearning methods, attacks and defenses.

Nice visualisations of different attack scenarios in Figure 1.

Extensive appendix with experimental details and additional comments on results.

**Weaknesses:**

My main concerns with this work are regarding its relevance and missing connections with the fine-tuning attacks community.

Authors claim that they introduce “the first comprehensive framework designed to formalize, evaluate, and benchmark unlearning attacks and defenses against LLMs”, however, the formalisation of each method is rather poor. Authors simply mention that the unlearning attack problem can be captured as finding a set of unlearning requests $D_{f}$, such that a set of constraints $C_i$ are met. While authors provide some examples of such constraints in lines 212 and 227, this is not enough as a formalization. For the framework to be general and useful, all the methods studied in this work should be formalised into this framework. As an example, for “Training-unlearning attack constraints.”, authors mention that “stealth” or “attack activation” constraints can be implemented. However, it’s completely unclear what these are or how they can be implemented.

A connection to fine-tuning attacks is missing. Unlearning methods can be thought of fine-tuning methods with the specific objective of “forgetting” training samples. For example, gradient ascent or other objectives can be used for unlearning [1,2]. It is well known that simply fine-tuning on a benign dataset can revert the alignment of LLMs [3]. Crafting the examples to use for fine-tuning to further elicit unsafe behaviours has recently gained traction [4]. If unlearning is just fine-tuning, how does your work fit within the vast harmful fine-tuning literature? Can’t you use any of the attacks and defenses described in [4] for unlearning?

Abuse of “\vspace{-x}” throughout the paper. The formatting of the paper has been broken by drastically reducing the margins of section headers and space before and after figures. The paper looks cluttered.

The paper heavily relies on the reader going back and forth to the appendix for details. Figure captions are rather short and many metrics and terms are deferred to the appendix. This makes it difficult to read and understand the paper.

**References**

[1] Zhang et al., Negative Preference Optimization: From Catastrophic Collapse to Effective Unlearning, COLM 2024

[2] Golatkar et al., Eternal Sunshine of the Spotless Net: Selective Forgetting in Deep Networks, CVPR 2020

[3] Qi et al., Fine-tuning Aligned Language Models Compromises Safety, Even When Users Do Not Intend To!, ICLR 2024

[4] Huang et al., Harmful Fine-tuning Attacks and Defenses for Large Language Models: A Survey, ArXiv 2024

**Questions:**

- Are you able to capture all of the cited methods within your framework?
- What are the key differences between unlearning attacks and the broader class of fine-tuning attacks?
- Why do we need a new framework specifically for unlearning attacks?

---

> ### Author Response · Authors · 2025-11-21
> **Response to Reviewer ddTi (Part 1/8)**
>
> **Dear Reviewer ddTi,**
>
> We would like to sincerely thank you for your valuable comments and the precious time and effort you devoted to reviewing our work. All of the raised comments are very helpful for us to improve our work. We have carefully integrated them into the revised version of the paper. Our point-by-point responses to the raised comments are provided below. We in the final version will include all of them and gratefully acknowledge the anonymous reviewers in the acknowledgments section.
>
> **Q1. Are there missing connections with the fine-tuning attacks community in this work? Does the paper consider the relevance of this work to the fine-tuning attacks?**
>
> **A1.** Thank you very much for the constructive comments. First, we would like to clarify that **in our initial submission, we have already included the connections and the relevance to fine-tuning attacks (please refer to the first paragraph of Section 11.2, Figure 7, Figure 6, Table 7, and  Figure 13 of the paper.)**. Specifically, in the first paragraph of Section 11.2 in our initial submission, we have discussed how unlearning attacks relate to malicious fine-tuning, highlighting unique threats of unlearning attacks during the unlearning stage. Below, we provide more detailed explanations for these included connections and relevance in Figure 7, Figure 6, Table 7, and  Figure 13  in our initial submission. Note that Figure 6 and Figure 13 from the initial submission now appear as Figure 8 and Figure 14 in the revised version.
>
> First, **in Figure 13 of our initial submission, we have examined fine-tuning attacks with benign data**. Specifically, we first fine-tuned the model on the Tofu dataset (a benign dataset) to obtain a fine-tuned model, after which we applied unlearning to a selected subset of the fine-tuned dataset. Here, we considered the sole unlearning attack scenario. For your convenience, we have provided the corresponding results in **Table 1** below. We adopted Mistral-7B-Instruct-v0.1 and Vicuna-7B-v1.5 models. We evaluated jailbreak attacks using the AdvBench dataset and measured the attack success rate (ASR) using both keyword-based and LLM-judge-based metrics. As shown in the table, the ASR under unlearning attacks increases even when the model has already undergone fine-tuning attacks (0% unlearning ratio). Moreover, as the unlearning ratio grows, the ASR continues to rise across both evaluation metrics. These results demonstrate that, even for models that have been fine-tuned on benign data, unlearning attacks can further deteriorate model robustness.
>
> Table 1: ASR (%) of fine-tuning attacks and unlearning attacks with various unlearning ratios on jailbreak performance.
> |Model|Metric|0%|1%|5%|10%|15%|20%|
> |:---|:---|:---|:---|:---|:---|:---|:---|
> |Mistral-7B-Instruct-v0.1|keyword-based ASR|86.00|86.00|82.00|100.00|100.00|100.00|
> |Mistral-7B-Instruct-v0.1|LLM-judge-based ASR|75.00|75.00|73.00|92.00|93.00|92.00|
> |Vicuna-7B-v1.5|keyword-based ASR|6.00|7.00|6.00|6.00|45.00|63.00|
> |Vicuna-7B-v1.5|LLM-judge-based ASR|4.00|6.00|5.00|6.00|3.00|22.00|
>
> Second, **in Figure 7 of our initial submission, we have examined how malicious unlearning attacks can be leveraged to activate hidden stealthy fine-tuning attacks**, thereby achieving both effectiveness and stealth. Specifically, in Figure 7, we considered the training–unlearning attack scenario. More specifically, we first performed the fine-tuning attacks by fine-tuning the pretrained model on a combination of malicious data and mitigation data. This fine-tuned model embeds the malicious behavior, but it remains latent and does not exhibit obvious harmful patterns at this fine-tuning stage. During the unlearning phase, portions of the mitigation data were removed, allowing the malicious behavior to surface. This mechanism differs fundamentally from traditional fine-tuning attacks: it remains stealthy and provides substantial flexibility and control. By adjusting unlearning requests, the attacker can finely regulate when the malicious behavior is activated, effectively overcoming the exposure risks typically associated with conventional fine-tuning attacks. For your convenience, we have moved corresponding results from **Figure 7** in the main manuscript into **Table 2** below. As shown, the harmful behavior remains effectively suppressed before unlearning but becomes pronounced once mitigation data are removed.

---

> ### Author Response · Authors · 2025-11-21
> **Response to Reviewer ddTi (Part 2/8)**
>
> Table 2: ASR (%) of training-unlearning attacks via hidden poisoning and informative poisoning methods under different unlearning methods and unlearning ratios.
> | Hidden poisoning | 5%   | 10%   | 15%   | 20%  | 25%  |
> | :---|:--- | :--- | :---| :--- | :--- |
> | GA     | 91.00 ± 3.61 | 95.33 ± 0.88 | 95.67 ± 0.88 | 96.33 ± 0.33 | 98.00 ± 1.15 |
> | GA+GD  | 83.33 ± 2.85 | 96.67 ± 0.88 | 94.33 ± 1.20 | 96.67 ± 1.20 | 97.33 ± 0.88 |
> | NPO    | 31.33 ± 2.91 | 29.67 ± 2.40 | 28.33 ± 3.18 | 30.00 ± 1.00 | 29.67 ± 0.88 |
> | NPO+GD | 35.33 ± 7.88 | 39.00 ± 4.36 | 42.33 ± 7.54 | 56.00 ± 2.52 | 77.33 ± 5.78 |
> |Informative poisoning|10%|20%|30%|40%|50%|
> | GA     | 4.00 ± 2.00 | 5.67 ± 1.76 | 5.00 ± 2.52   | 29.67 ± 24.69 | 10.00 ± 7.51  |
> | GA+GD  | 6.33 ± 4.91 | 0.33 ± 0.33 | 19.67 ± 12.73 | 0.67 ± 0.67   | 21.00 ± 20.50 |
> | NPO    | 0.33 ± 0.33 | 0.00 ± 0.00 | 0.00 ± 0.00   | 0.00 ± 0.00   | 0.00 ± 0.00   |
> | NPO+GD | 1.00 ± 0.00 | 0.33 ± 0.33 | 13.33 ± 6.17  | 23.67 ± 11.61 | 18.00 ± 7.64  |
>
> Third, **in Table 7 of our initial submission, we have also examined how malicious unlearning attacks can be leveraged to activate hidden fine-tuning attacks**, thereby achieving both effectiveness and stealth. Specifically, in this experiment, we first performed the fine-tuning attacks by fine-tuning the pretrained model on a combination of malicious data and mitigation data. This fine-tuned model embeds the malicious behavior, but it remains latent and does not exhibit obvious harmful patterns at this stage. During the unlearning phase, portions of the mitigation data were removed, allowing the malicious behavior to surface. This mechanism differs fundamentally from traditional fine-tuning attacks: it remains stealthy and provides substantial flexibility and control. For your convenience, we have moved corresponding results from **Table 7** in the main manuscript into **Table 3** below. The results suggest that the malicious behavior is effectively concealed before unlearning, but becomes evident when the mitigation data are removed.
>
> Table 3: ASR (%) of training-unlearning attacks via hidden poisoning method using different unlearning methods on different models across unlearning ratios.
> | Model | Unlearning ratio | GA | GA+GD | NPO | NPO+GD |
> | :--- | :--- | :--- | :--- | :--- | :--- |
> |Qwen2.5-1.5B| 0% | $28.33 \pm 1.33$ | $28.33 \pm 1.33$ | $28.33 \pm 1.33$ | $28.33 \pm 1.33$ |
> | | 5% | $13.00\pm5.51 $ | $27.00\pm11.06 $ | $40.00\pm1.73 $ | $51.00\pm6.66 $ |
> | | 10% | $83.33\pm7.69$ | $79.33\pm5.36 $ | $46.33\pm3.48 $ | $64.33\pm5.93 $ |
> | | 15% | $96.00\pm2.00 $ | $94.33\pm2.40 $ | $51.67\pm2.33$ | $76.67\pm0.67 $ |
> | | 20% | $99.33\pm0.67$ | $97.33\pm0.88 $ | $52.00\pm1.53 $ | $79.33\pm3.93 $ |
> | | 25% | $99.67\pm0.33$ | $97.67\pm1.20 $ | $51.33\pm2.33 $ | $87.67\pm0.33 $ |
> |Qwen2.5-3B| 0% | $0.67\pm0.33$ | $0.67\pm0.33$ | $0.67\pm0.33$ | $0.67\pm0.33$ |
> | | 5% | $31.67\pm3.18$ | $8.67\pm4.41$ | $0.67\pm0.33$ | $0.67\pm0.33$ |
> | | 10% | $87.67\pm1.33$ | $86.67\pm1.86$ | $0.67\pm0.33$ | $ 0.67\pm0.33$ |
> | | 15% | $94.00\pm1.15 $ | $95.67\pm1.20 $ | $0.67\pm0.33$ | $3.33\pm1.45 $ |
> | | 20% | $98.33\pm0.88 $ | $95.33\pm2.03 $ | $0.67\pm0.33$ | $8.00\pm1.53 $ |
> | | 25% | $99.67\pm0.33$ | $97.33\pm0.88 $ | $2.33\pm0.88$ | $28.67\pm1.86$ |
> |Qwen2.5-7B| 0% | $8.67\pm1.33$ | $8.67\pm1.33$ | $8.67\pm1.33$ | $8.67\pm1.33$ |
> | | 5% | $76.00\pm1.53 $ | $75.33\pm2.33 $ | $28.33\pm2.73 $ | $36.00\pm5.77 $ |
> | | 10% | $93.33\pm4.18 $ | $89.00\pm4.04 $ | $62.00\pm2.65$ | $61.00\pm7.51 $ |
> | | 15% | $96.67\pm2.33$ | $94.33\pm1.76 $ | $81.67\pm2.33$ | $49.00\pm7.64 $ |
> | | 20% | $100.00\pm0.00$ | $98.00\pm1.00 $ | $87.33\pm1.76$ | $52.33\pm2.19 $ |
> | | 25% | $100.00\pm0.00$ | $97.67\pm0.88 $ | $92.00\pm2.00 $ | $70.00\pm4.58 $ |

---

> ### Author Response · Authors · 2025-11-21
> **Response to Reviewer ddTi (Part 3/8)**
>
> Fourth, **in Figure 6 of our initial submission, we have examined how malicious unlearning attacks can be leveraged to activate hidden fine-tuning based backdoor attacks**, thereby achieving both effectiveness and stealth. Specifically, in this training–unlearning-inference attack scenario, we first performed the fine-tuning attacks by fine-tuning the pretrained model on a composite dataset consisting of clean training dataset, trigger-poisoned malicious dataset, and mitigation dataset. This fine-tuned model contains an embedded backdoor, but at this stage the backdoor remains dormant and does not exhibit trigger-activated behavior. During the later unlearning phase, portions of the mitigation data were removed, enabling the trigger-activated malicious behavior to emerge. This mechanism differs fundamentally from traditional fine-tuning based backdoor attacks: it maintains stealth and provides substantial flexibility and controllability. By adjusting unlearning requests, the adversary can finely determine when and how the backdoor is activated, effectively mitigating the exposure risks typically associated with conventional backdoor techniques. For your convenience, we have moved corresponding results from **Figure 6** in the main manuscript into **Table 4** below. The evidence demonstrates that the backdoor behavior stays suppressed before unlearning, but is triggered once the mitigation data are removed.
>
>
> Table 4: ASR of training-unlearning-inference attacks across different unlearning ratios and LLMs.
> |Model|0%|1.43%|2.86%|4.29%|5.71%|7.14%|
> |:---|:---|:---|:---|:---|:---|:---|
> |Llama-2-7B (w/ / w/o) |  |  |  |  |  |  |
> |BadNets|0.04 / 0.08|0.58 / 0.02|0.84 / 0.04|0.44 / 0.04|0.96 / 0.10|1.00 / 0.04|
> |UBA-Inf|0.14 / 0.02|0.78 / 0.04|0.80 / 0.04|0.56 / 0.02|1.00 / 0.08|1.00 / 0.06|
> |ReVeil|0.06 / 0.06|0.20 / 0.04|0.36 / 0.02|0.04 / 0.02| 0.72 / 0.04 |0.98 / 0.06|
> |UBA-Inf + ReVeil|0.04 / 0.02|0.62 / 0.06|0.16 / 0.06|0.42 / 0.02|1.00 / 0.08|0.98 / 0.08|
> | Mistral-7B-Instruct  (w/ / w/o)   |  |  |  |  |  |  |
> |BadNets|0.62 / 0.06|0.86 / 0.02|0.58 / 0.04|0.76 / 0.00|0.68 / 0.02|1.00 / 0.02|
> |UBA-Inf|0.28 / 0.04|0.86 / 0.02|0.62 / 0.04|0.50 / 0.04|0.78 / 0.00|1.00 / 0.08|
> |ReVeil|0.14 / 0.06|0.10 /0.04|0.10 / 0.04|0.02 / 0.02|0.14 / 0.00   |1.00 / 0.00|
> |UBA-Inf + ReVeil|0.60 / 0.00|0.76 / 0.04|0.70 / 0.04|0.64 / 0.02   |0.68 / 0.02|1.00 / 0.04|
>
>
>
> Lastly, we would like to clarify that unlearning attacks are orthogonal to traditional attacks, including data poisoning attacks (during training), fine-tuning attacks (that happen during the fine-tuning stage and can also be viewed as a form of data poisoning attacks), and adversarial attacks (during inference). In the initial submission, we have also clarified the uniqueness of unlearning attacks and their differences with traditional security attacks. These attack types can individually and independently operate at different stages of the model lifecycle and exploit different mechanisms. Therefore, **unlearning attacks are orthogonal to traditional threats, and can introduce additional and distinct attack surfaces** that must be considered separately when studying the overall security and robustness of large language models.

---

> ### Author Response · Authors · 2025-11-21
> **Response to Reviewer ddTi (Part 4/8)**
>
> **Q2. Clarifying the formalisation of each method. Do the authors simply mention that the unlearning attack problem can be captured as finding a set of unlearning requests $D_f$, such that a set of constraints are met?**
>
> **A2.** Thank you very much for the valuable comments. First, we would like to clarify that **due to space limitations, in our initial submission, the formal formalisation of the unlearning attacks and defenses are deferred to Appendix (see Section 9.2, Section 10.2, Section 11.2, Section 12.2, Section 13.2, Section 14.2, and Section 15.2)**. These appendix sections provide complete formulations, challenges, equations, optimization objectives, and algorithmic procedures for the methods discussed in the main paper.
>
> Additionally, as an example illustration, for your convenience, we clarify the included formalisation of our methods in Section 9.2 of the initial submission. Specifically, *Section 9.2* provides the full formalization of sole unlearning attacks. To illustrate the level of formal rigor, we use the exact sole unlearning attack as an example. In *Equation 11*, we present a constrained optimization framework that captures the unique structure and goals of this attack scenario. Specifically, the first constraint $\mathcal{C}\_{1}$ ​encodes the attacker’s objective of inducing strong performance degradation on the unlearned model, while the second constraint $\mathcal{C}\_{2}$ enforces the exact unlearning requirement, i.e., that an exact subset of training samples must be removed. The third constraint specifies that the unlearned model $\theta\^u$ must result from executing the required unlearning algorithm $\mathcal{U}$. However, this initial formulation introduces challenges and is difficult to optimize directly. To address this, we reformulate the problem into the bi-level optimization structure in *Equation 12*, where the upper-level objective represents the attack goal, and the lower-level problem captures the exact unlearning process via discrete indication parameters. Even in this bi-level form, the optimization remains challenging due to its computational intractability. To make the problem solvable in practice, we adopt a gradient-matching technique that approximates the exact bi-level objective by aligning the gradient of the unlearning loss with that of the target attack loss. This transformation enables us to convert the bi-level problem into an efficiently solvable single-level optimization, presented in *Equation 13 and Equation 14*. As a result, the attack becomes computationally feasible and scalable. Finally, we also provide the complete algorithmic pipeline for this attack in *Algorithm 1*.  **The complete formalizations for the remaining attack scenarios are also fully detailed in their respective Appendix sections. These comprehensive formulations, together with the articulated challenges, equations, optimization objectives, and algorithmic procedures, collectively demonstrate that our framework provides a rigorous and comprehensive basis for formalizing, evaluating, and benchmarking unlearning attacks and defenses for large language models.**
>
> Further, we would like to clarify that the unlearning attack problem is inherently characterized by the unique nature of unlearning itself. Unlike traditional attacks, unlearning attacks aim to craft a set of malicious unlearning data $D_f$ such that the resulting unlearned model exhibits undesired behaviors. Therefore, capturing unlearning attacks as a problem of finding a set of malicious unlearning requests $D_f$ reflects the fundamental structure of the problem. At the same time, the generation of malicious unlearning data must obey a set of constraints that stem from practical unlearning settings, attacker capabilities, and the specific unlearning mechanisms being exploited. These constraints are explicitly formalized in Appendix (e.g., constraints on discrete selection of unlearning data, attack stealthiness, and other involved stages and data), and each method’s differences are reflected through the specific constraints they enforce.

---

> ### Author Response · Authors · 2025-11-21
> **Response to Reviewer ddTi (Part 5/8)**
>
> **Q3. 3.1) Do the authors only provide some examples of such constraints in lines 212 and 227 for a formalization? 3.2) Is the framework general and useful such that all the methods studied in this work can be formalised into this framework?**
>
> **A3.** Thank you very much for the insightful comment. Firstly, we would like to clarify that the examples provided in Lines 212 and 227 are intended to illustrate the potential constraints in the sole unlearning attack setting, rather than to serve as the full formalization. We also would like to clarify that **due to space limitations, we have deferred the full formalisation (including the attack frameworks, attack constraints, optimization, and algorithms) into Appendix (see Section 9.2)**. Specifically, *Section 9.2* presents the attack frameworks in *Sections 9.2.1 and 9.2.2*, the corresponding attack formalizations with constraints (*Equations 11, 15, 18, 20*), reformulated objectives (*Equations 12, 16, 19, 21*) , the optimization objectives (*Equations 13, 14, 17, 22*), and the detailed algorithms (*Algorithms 1, 2, 3*).
>
> Additionally, we would like to emphasize that our proposed framework is general, and all the methods studied in this work can indeed be formalized within this unified framework. Specifically, all the methods aim to craft malicious unlearning requests under the machine unlearning paradigm, which naturally formulates the problem as identifying malicious unlearning data that induces undesirable behavior in the resulting unlearned model. Their distinctions are reflected through the different sets of enforced constraints, which are fully detailed in Appendix (please refer to Section 9.2, Section 10.2, Section 11.2, Section 12.2, Section 13.2, and Section 14.2).
>
>
> **Q4. Clarify what these "stealth" or "attack activation" constraints are or how they can be implemented for “Training-unlearning attack constraints”.**
>
> **A4.** Thank you very much for the valuable comments. We first would like to clarify that **due to space limitations, in our initial submission, the complete implementation of the “stealth” and “attack activation” constraints for “Training-unlearning attack” has been provided in Section 11.2 of Appendix**. Specifically, in *Equation 29* of the initial submission, we have defined the stealth constraint $\mathcal{C}\_{1}$ as requiring the adversarial loss of the fine-tuned model $\theta^t$ trained on the carefully crafted points $\mathcal{D}\_{cr}$ remains below a predefined threshold, ensuring that the carefully crafted points added to the training dataset has a minimal impact on the model performance. On the other hand, we have defined the attack activation constraint $\mathcal{C}\_{2}$ as requiring that the adversarial loss of the unlearned model is below a predefined threshold. This indicates that the removal of a subset of the carefully crafted points $\mathcal{D}\_{cr}$ activates the effect of poisoning attacks in the training-unlearning stage, leading to significant model degradation in the resulting unlearned model.  **Importantly, these constraints are not only defined mathematically but are also reflected in the attack workflow.** As illustrated in *Example 11.1*, the mitigation data are constructed using the same prompt as the poisoning data but with opposite answers, which ensures that fine-tuning on these points satisfies the stealth constraint by preserving the model’s normal behavior. Furthermore, *Figure 26* provides a visual example showing how unlearning a targeted subset triggers the poisoning effect, thereby satisfying the attack-activation constraint and leading to substantial degradation in the unlearned model.

---

> ### Author Response · Authors · 2025-11-21
> **Response to Reviewer ddTi (Part 6/8)**
>
> **Q5. Can you use any of the attacks and defenses described in [4] for unlearning?**
>
> **A5.** Thank you very much for the thoughtful comments. Following your suggestions, we have adopted the attacks and defenses described in [4] for our unlearning settings. Specifically, we conducted new experiments to demonstrate that unlearning attacks can both enhance and degrade the effectiveness of fine-tuning attacks with and without defenses. More specifically, we used the fine-tuning attack Shadow Alignment [5] from [4] to break the safety alignment of LLMs and induce harmful outputs. Based on the sole unlearning attack setting on the Tofu dataset, we further fine-tuned one hundred harmful examples from [5] and then applied unlearning attacks using the GA unlearning method with a 5% unlearning ratio. We evaluated the Llama2-13B-Chat, Qwen2.5-1.5B-Instruct, and Qwen2.5-7B-Instruct models, using the CoNa harmful dataset from [5] and the LLM-judge-based ASR metric. For defenses, we adopted the method SafeInstr [6] from [4], which incorporates safety alignment data  (for one hundred)  into the fine-tuning process to defend against fine-tuning attacks. Table 5 reports the results of fine-tuning attacks with and without defenses under unlearning attacks. As shown, fine-tuning attacks alone substantially increase harmful content, but this effect is largely mitigated by the defense method. However, with different unlearning attack objectives, unlearning can either enhance or degrade the effectiveness of fine-tuning attacks in both defended and undefended fine-tuning attack settings. The corresponding fine-tuning attack and defense results with unlearning have also been included in Figure 19 of Appendix Section 9.3.2.
>
> Table 5: ASR (%) of fine-tuning attacks with and without defenses under unlearning attacks.
> |Experiment setting|Llama2-13B-Chat|Qwen2.5-1.5B-Instruct|Qwen2.5-7B-Instruct|
> |:---|:---|:---|:---|
> |Baseline|0.00|9.00|11.00|
> |Fine-tuning attack [5]|67.00|37.00|52.00|
> |Unlearning attack enhances fine-tuning attack|69.00|44.00|63.00|
> |Unlearning attack degrades fine-tuning attack|1.00|25.00|11.00|
> |Defense against fine-tuning attack [6]|24.00|25.00|13.00|
> |Unlearning attack enhances defended fine-tuned model|54.00|28.00|21.00|
> |Unlearning attack degrades defended fine-tuned model|9.00|8.00|4.00|
>
>
>
>
> **Q6. Addressing the formatting of the paper due to the use of “\vspace{-x}” for large margin spaces.**
>
> **A6.** Thank you very much for the helpful suggestion. First, we would like to clarify that due to space limitations and the diverse settings in the paper, we have adopted it to include more details for the considered settings. Additionally, following your suggestion, we have revised the formatting of the paper in the revised version for large margin spaces.
>
>
> **Q7. Why looking for details by going to the Appendix? Why are many metrics and terms deferred to the Appendix and figure captions are short?**
>
> **A7.** Thank you very much for the valuable suggestions. First, we would like to clarify that due to the page and space limitations and our considered diverse settings, we deferred some details (including some metrics and terms) to Appendix. At the same time, we have made extensive efforts to include more complete and comprehensive details in Appendix. This can balance focus in the main text with more detailed completeness in Appendix sections. Additionally, following your suggestions, we have provided longer and more detailed figure captions in the revised manuscript (Figures 2-8, and 10-12), and with more clear definitions of the metrics and terms used in the figures (the third paragraph in Section 4.1 of the main paper). Further, for the evaluation metrics, we have added the metric of detection accuracy to the main manuscript. This ensures that all metrics used in our experiments are now clearly defined within the main manuscript.
>
>
> **Q8. Are you able to capture all of the cited methods within your framework?**
>
> **A8.** Thank you very much for the constructive comments. We would like to clarify that for all of the cited unlearning attack methods in the framework (see Section 3 of the paper), which are also summarized and categorized in Table 1, we have captured all of them within our framework. Additionally, for all of the cited unlearning defenses in the framework (see Section 3 of the paper), we have also captured all of them within our framework.

---

> ### Author Response · Authors · 2025-11-21
> **Response to Reviewer ddTi (Part 7/8)**
>
> **Q9. Clarifying the key differences between unlearning attacks and the broader class of fine-tuning attacks?**
>
> **A9.** Thank you very much for the valuable comment. First, we would like to clarify that **as discussed in the first paragraph of Section 11.2 in our initial submission, unlearning attacks and fine-tuning attacks have different attack timings**. Unlearning attacks happen during the unlearning stage, where attackers manipulate the model’s parameters via unlearning some training samples. In contrast, fine-tuning attacks happen during the fine-tuning stage, where attackers inject new fine-tuning data to manipulate the model’s parameters. Notably, the two different stages could happen independently. For example, for a pre-trained model, in the real-world deployment, it can have different fine-tuning stages to adapt to new tasks, and it can also involve different unlearning stages for unlearning undesired knowledge. This highlights that to achieve robust and trustworthy model performance, **researchers and practitioners must account for vulnerabilities arising from multiple lifecycle stages, including the fine-tuning stage, and the unlearning stage**.
>
> Additionally, they have different mechanisms, goals, and constraints. Unlearning attacks aim to remove the target requested unlearning data (that is a subset of the original training data). Fine-tuning attacks typically use new data to inject new behaviors into the model. For unlearning attacks, they aim to adopt existing machine unlearning methods to induce harmful behaviors in the resulting unlearned model, while fine-tuning attacks over the new fine-tuning data adopt fine-tuning optimizations (instead of unlearning algorithms). They also have different constraints. For example, unlearning attacks generally operate under the constraint that the malicious unlearning requests must correspond to data that originate from the model’s original training set, since unlearning procedures are designed to remove previously used training samples. In contrast, fine-tuning attacks are not restricted in this way, since attackers can introduce arbitrary new data during the fine-tuning stage.
>
> Further, it is important to highlight that **unlearning attacks and fine-tuning attacks reveal vulnerabilities that emerge at different operational stages of the model pipeline**. Notably, based on the above discussions, unlearning attacks and fine-tuning attacks, **they could also interplay with each other**. Hence, both attack classes are indispensable for a complete understanding of model vulnerabilities and robustness at different operational stages of the model pipeline.
>
>
> **Q10. 10.1) How does your work fit within the vast harmful fine-tuning literature if unlearning is just fine-tuning? 10.2) Discuss the case where gradient ascent can be used for unlearning [1,2]. 10.3) Discuss the case where fine-tuning on benign data [3]. 10.4) Discuss the case of crafting fine-tuning data for unsafe behaviors [4].**
>
> **A10.** Thank you very much for the constructive comments. Firstly, we would like to clarify that unlearning attacks are orthogonal to traditional attacks, including data poisoning attacks (during training), fine-tuning attacks (that happen during the fine-tuning stage and can also be viewed as a form of data poisoning attacks), and adversarial attacks (during inference). These attack types can individually and independently operate at different stages of the model lifecycle and exploit different mechanisms. Therefore, **unlearning attacks are orthogonal to traditional threats, and can introduce additional and distinct attack surfaces** that must be considered separately when studying the overall security and robustness of large language models. In the initial submission, we have also clarified the uniqueness of unlearning attacks and their differences with traditional security attacks.
>
> Additionally, we would like to clarify that **unlearning attacks can open a new door for achieving fine-tuning attacks during the unlearning stage (instead of the fine-tuning stage)**. This could happen when attackers launch unlearning attacks via the unlearning tools (e.g., gradient ascent based unlearning methods [1,2]) to achieve harmful fine-tuning. This further highlights the importance of studying unlearning attacks in our work, since **unlearning attacks could interplay with traditional security attacks (including harmful fine-tuning)** and complicate the model robustness and vulnerabilities.

---

> ### Author Response · Authors · 2025-11-21
> **Response to Reviewer ddTi (Part 8/8)**
>
> Further, we would like to clarify that **in Figure 13 of our initial submission, we have examined fine-tuning attacks with benign data**. Specifically, we first fine-tuned the model on the Tofu dataset (that is a benign dataset) to obtain a fine-tuned model, after which we applied unlearning to a selected subset of the fine-tuned dataset for malicious unlearning attacks. The obtained experimental results show that the attack success rate under unlearning attacks increases even when the model has already been fine-tuned on the benign data [3]. Note that Figure 13 from the initial submission now appears as Figure 14 in the revised version.
>
> Lastly, in Table 6 above, we have conducted experiments to show that for the case of crafting fine-tuning data for unsafe behaviors [4], **unlearning attacks can also enhance or reduce the impact of fine-tuning attacks** with and without defenses. This further reveals the complex and non-trivial interactions between unlearning attacks and fine-tuning attacks, and underscores the importance of our work in guiding and inspiring more innovative research studies in this direction.
>
>
> **Q11. Why do we need a new framework specifically for unlearning attacks?**
>
> **A11.** Thank you very much for the precious comment. First, we would like to clarify that, as discussed in the Introduction (the second paragraph of Section 1), LLM unlearning introduces an additional interaction surface within the system. This new interface enables adversaries to exploit the unlearning mechanism by submitting malicious unlearning requests, which can induce undesirable behaviors in the resulting unlearned models. However, there is no existing work studying the robustness and safety performance of LLMs under unlearning attacks. To bridge this gap, in this paper, we propose a new framework for LLM unlearning attacks. This framework is necessary since: (i) Unlearning attacks form a fundamentally new class of threats, since they exploit the unlearning paradigm and the new unlearning stage. (ii) Our unified framework formalizes, categorizes, and compares unlearning attacks within the new machine unlearning paradigm. In this way, we can have an overall understanding of the connections and relationships among different unlearning attacks. (iii) Our new framework provides a principled way to benchmark and evaluate unlearning attacks. (iv) Our new framework can help to clarify the distinctions between unlearning attacks and traditional security threats. (v) Our new framework can help to inspire more innovative explorations in the subsequent study, and can guide researchers and practitioners toward more comprehensive security analyses of model vulnerabilities in large language models.
>
>
> References
>
> [1] Zhang et al., Negative Preference Optimization: From Catastrophic Collapse to Effective Unlearning, COLM 2024
>
> [2] Golatkar et al., Eternal Sunshine of the Spotless Net: Selective Forgetting in Deep Networks, CVPR 2020
>
> [3] Qi et al., Fine-tuning Aligned Language Models Compromises Safety, Even When Users Do Not Intend To!, ICLR 2024
>
> [4] Huang et al., Harmful Fine-tuning Attacks and Defenses for Large Language Models: A Survey, ArXiv 2024
>
> [5] ​​Shadow Alignment: The Ease of Subverting Safely-Aligned Language Models, arXiv 2023
>
> [6] Safety-Tuned LLaMAs: Lessons From Improving the Safety of Large Language Models that Follow Instructions, ICLR 2024
>
>
> **Sincerely yours,**
>
> **Authors of Paper 13906**

---

> > ### Comment · Reviewer_ddTi · 2025-11-24
> >
> > Thanks to the authors for the long and detailed response.
> >
> > - **P1:** “we have already included the connections and the relevance to fine-tuning attacks (please refer to the first paragraph of Section 11.2, Figure 7, Figure 6, Table 7, and Figure 13 of the paper.”
> >
> >
> > In section 11.2 you do not cite **Any** malicious fine-tuning attack reference.
> > Figure 13 indeed contains fine-tuned models (no unlearning) when the unlearning ratio is 0%, however you do not include any citations to reference [1] ([3] in the original review) or any other fine-tuning attack reference.
> >
> > In Figure 7 and the corresponding Table 7, the methods you evaluate are unlearning based, there is no link or comparison to malicious fine-tuning methods. The same applies to Figure 6, where you do not even cite the original BadNets paper [2] used by the unlearning method [3] you evaluate! Instead of calling the method you reference “BadNets”, you should call it BAU as denoted by [3].
> >
> > All in all, there is **no reference** to malicious fine-tuning attacks in the paper.
> >
> > - **P2:** On the formalization framework (Q2-Q4)
> >
> > Thanks to the authors for pointing out to your formalization of existing methods in sections 9.2, 10.2, 11.2, 12.2, 13.2, 14.2, and 15.2.
> >
> > However, if one of your core contributions is to formalize unlearning attacks and defenses, as marketed in your title, abstract and introduction, this should be included explicitly in the main text, not hidden in a 50+ page appendix.
> >
> > - **P3:** On the formatting infringements
> >
> > According to the style guide in https://iclr.cc/Conferences/2026/AuthorGuide:
> >
> > “Authors are required to use the ICLR LATEX style files obtainable at the ICLR website. Please make
> > sure you use the current files and not previous versions. Tweaking the style files may be grounds for
> > rejection.”
> >
> > Even after extending to 10 pages during the rebuttal, every margin after a section or subsection header has been clearly tweaked. “we have adopted it to include more details” is not a valid argument.
> >
> > **Final remarks**
> >
> > The paper needs a major revision, focusing on 1) clearly stating how different unlearning attacks and defenses are formalized in the main paper **in the main text** 2) Clearly differentiating from fine-tuning attacks and justifying the necessity of a new framework **in the main text** 3) adhering to the formatting instructions.
> >
> > **References**
> >
> > [1] Qi et al., Fine-tuning Aligned Language Models Compromises Safety, Even When Users Do Not Intend To!, ICLR 2024
> >
> > [2] Gu et al., BadNets: Evaluating Backdooring Attacks on Deep Neural Networks, IEEE Access 2019
> >
> > [3] Zhang et al., Exploiting machine unlearning for backdoor attacks in deep learning system, ArXiv 2023

---

> > > ### Author Response · Authors · 2025-11-27
> > > **Thank you very much for the prompt feedback (Part 1/6)**
> > >
> > > **Dear Reviewer ddTi,**
> > >
> > > We sincerely appreciate your valuable feedback and helpful suggestions. All of the raised comments are very helpful for us to improve our work. We have also carefully integrated the provided suggestions into the revised version of the paper. Our point-by-point responses to the raised comments are provided below.
> > >
> > >
> > > **Q1. Do the authors tweak the style files?**
> > >
> > > **A1.** Thank you very much for your valuable comments. We would like to clarify that **our submission strictly adheres to the official ICLR LATEX style files (including natbib.sty, iclr2026_conference.sty, and fancyhdr.sty)** obtainable on the conference website (https://iclr.cc/Conferences/2026/AuthorGuide), and we did not modify any LATEX style files. We also would like to clarify that we have not made any changes to the style files.
> > >
> > >
> > > **Q2. Discussions on margin after a section or subsection header.**
> > >
> > > **A2.** Thank you very much for the valuable comments. First, we would like to clarify that we have strictly followed the official style files and the templates obtained from the ICLR website to prepare our submission. We want to clarify that we did not modify any LATEX style files. Additionally, from the official template (iclr2026_conference.tex) that we downloaded from the conference website (https://iclr.cc/Conferences/2026/AuthorGuide) to compile the pdf submission, we noticed that **this official template (iclr2026_conference.tex) includes some uses of “\vspace{}” (e.g., Lines 254, 271, 289, 310, 329, and 348 in iclr2026_conference.tex) to adjust the margin** accordingly. Further, following your suggestion, we have carefully reviewed the author guidelines for the conference camera ready version preparation, we found no instruction prohibiting the use of the standard “\vspace{}” command in the paper submission. Lastly, following your suggestion, we have deleted the “\vspace{}” for the margins after these heads in the revised version.
> > >
> > >
> > > **Q3. Thanks to the authors for pointing out to your formalizations in Section 9.2 of Appendix, Section 10.2 of Appendix, Section 11.2 of Appendix, Section 12.2 of Appendix, Section 13.2 of Appendix, Section 14.2 of Appendix, and Section 15.2 of Appendix.**
> > >
> > > **A3.** Thank you very much for the precious comments. We are very pleased to hear that we have clarified that in our initial submission, we have included these formalizations in Section 9.2, Section 10.2, Section 11.2, Section 12.2, Section 13.2, Section 14.2, and Section 15.2. We would like to sincerely thank you for this feedback.

---

> > > ### Author Response · Authors · 2025-11-27
> > > **Thank you very much for the prompt feedback (Part 2/6)**
> > >
> > > **Q4. Please explicitly in the main text include the following 7 sections in Appendix: method formalizations in Section 9.2 of Appendix, Section 10.2 of Appendix, Section 11.2 of Appendix, Section 12.2 of Appendix, Section 13.2 of Appendix, Section 14.2 of Appendix, and Section 15.2 of Appendix.**
> > >
> > > **A4.** Thank you very much for the constructive comments. First, we would like to clarify that to provide more details for the framework in the main text, we have devoted extensive efforts to provide more detailed method formalizations in Section 9.2, Section 10.2, Section 11.2, Section 12.2, Section 13.2, Section 14.2, and Section 15.2. To assist readers in locating these materials efficiently, we have also included an outline on page 19 to provide a quick index to these sections. These sections are organized in a structured and clear manner. We also want to clarify that we did not hide them in Appendix; instead, we clearly structured them to provide more details while following the strict page requirement of the main text.
> > >
> > > Additionally, we would like to clarify that due to the strict paper page requirement (9 page requirement for the main text in the initial submission), appendix (50+ pages) is one of our extensively dedicated efforts to provide more details and clarifications for the comprehensive benchmark framework in our work.
> > >
> > > Moreover, we would like to clarify that the detailed method formalizations in **these 7 Appendix sections span around 19 pages in total** (i.e., Section 9.2 of Appendix, Section 10.2 of Appendix, Section 11.2 of Appendix, Section 12.2 of Appendix, Section 13.2 of Appendix, Section 14.2 of Appendix, and Section 15.2 of Appendix). This also shows our dedicated extensive efforts in providing more details in Appendix.
> > >
> > > Lastly, we would like to clarify that in Eq. (1) of the main text, we have provided a general framework, and all the attack methods studied in this work can indeed be formalized within this unified framework. Specifically, all the methods aim to craft malicious unlearning requests under the machine unlearning paradigm, which naturally formulates the problem as identifying malicious unlearning data that induces undesirable behavior in the resulting unlearned model. Their distinctions are reflected through the different sets of enforced constraints, which are fully detailed in Appendix (see sections 9.2, 10.2, 11.2, 12.2, 13.2, 14.2, and 15.2). While adhering to the strict 10-page limit for the ICLR revision, we have also followed your suggestions and moved the formalization in Eq. (11) in Section 9.2 to the main text of the revised version. Additionally, we have clearly indicated in the main text where these more details can be found in Appendix.
> > >
> > >
> > >
> > > **Q5. Clarifications on Figure 7 and the corresponding Table 7 with respect to malicious fine-tuning, and Figure 6.**
> > >
> > > **A5.** Thank you very much for your valuable comments. In Figure 7, the unlearning attacks clearly link to malicious fine-tuning, as unlearning attacks in this setting are designed to activate the stealthy malicious fine-tuning attacks embedded during the initial fine-tuning stage. In this setting, we first performed the fine-tuning attacks by fine-tuning the pretrained model on a combination of malicious data and mitigation data. This fine-tuned model embeds the malicious behavior, but it remains latent and does not exhibit obvious harmful patterns at this fine-tuning stage. The unlearning process gradually removes the mitigation data and thereby activates the underlying malicious fine-tuning behavior. The links and comparisons are as follows: When no mitigation data has been removed, which is the case of unlearning 0% of the whole fine-tuning dataset, the attack performance reflects how well the mitigation data masks the malicious fine-tuning behavior. When the unlearning ratio increases, the masking effect diminishes, and the latent malicious fine-tuning behavior progressively resurfaces. When all mitigation data has been removed, which is the case of unlearning 25% of the whole fine-tuning dataset for hidden poisoning attacks,  the attack reduces to a standard malicious fine-tuning attack. In summary, our unlearning attacks expose a new and intricate attack surface: **unlearning attacks not only operate as independent and new attack relative to malicious fine-tuning attacks, but also interact with malicious fine-tuning attacks by activating stealthy and hidden fine-tuning attacks** (that were embedded in the fine-tuning stage). For your convenience, we have moved the results of Figure 7 into Table 1 below.

---

> > > ### Author Response · Authors · 2025-11-27
> > > **Thank you very much for the prompt feedback (Part 3/6)**
> > >
> > > Table 1: ASR (%) of training-unlearning attacks via hidden poisoning and informative poisoning methods under different unlearning methods and unlearning ratios.
> > > | Hidden poisoning | 0%   | 5%   | 10%   | 15%   | 20%  | 25%  |
> > > | :---|:--- | :--- | :---| :--- | :--- |:--- |
> > > | GA     | 34.00 ± 1.73 |91.00 ± 3.61 | 95.33 ± 0.88 | 95.67 ± 0.88 | 96.33 ± 0.33 | 98.00 ± 1.15 |
> > > | GA+GD  | 34.00 ± 1.73 |83.33 ± 2.85 | 96.67 ± 0.88 | 94.33 ± 1.20 | 96.67 ± 1.20 | 97.33 ± 0.88 |
> > > | NPO    | 34.00 ± 1.73 |31.33 ± 2.91 | 29.67 ± 2.40 | 28.33 ± 3.18 | 30.00 ± 1.00 | 29.67 ± 0.88 |
> > > | NPO+GD | 34.00 ± 1.73 |35.33 ± 7.88 | 39.00 ± 4.36 | 42.33 ± 7.54 | 56.00 ± 2.52 | 77.33 ± 5.78 |
> > > |Informative poisoning|0%|10%|20%|30%|40%|50%|
> > > | GA     | 0.33 ± 0.33 |4.00 ± 2.00 | 5.67 ± 1.76 | 5.00 ± 2.52   | 29.67 ± 24.69 | 10.00 ± 7.51  |
> > > | GA+GD  | 0.33 ± 0.33 |6.33 ± 4.91 | 0.33 ± 0.33 | 19.67 ± 12.73 | 0.67 ± 0.67   | 21.00 ± 20.50 |
> > > | NPO    | 0.33 ± 0.33 |0.33 ± 0.33 | 0.00 ± 0.00 | 0.00 ± 0.00   | 0.00 ± 0.00   | 0.00 ± 0.00   |
> > > | NPO+GD | 0.33 ± 0.33 |1.00 ± 0.00 | 0.33 ± 0.33 | 13.33 ± 6.17  | 23.67 ± 11.61 | 18.00 ± 7.64  |
> > >
> > > In Table 7, the unlearning attacks clearly link to malicious fine-tuning, as unlearning attacks in this setting are designed to activate the stealthy malicious fine-tuning attacks embedded during the initial fine-tuning stage. In this setting, we first performed the fine-tuning attacks by fine-tuning the pretrained model on a combination of malicious data and mitigation data. This fine-tuned model embeds the malicious behavior, but it remains latent and does not exhibit obvious harmful patterns at this stage. The unlearning process gradually removes the mitigation data and thereby reveals the underlying malicious fine-tuning behavior. The links and comparisons are as follows: When no mitigation data has been removed, which is the case of unlearning 0% of the whole fine-tuning dataset, the attack performance reflects how well the mitigation data masks the malicious fine-tuning behavior. When the unlearning ratio increases, the masking effect diminishes, and the latent malicious fine-tuning behavior progressively resurfaces. When all mitigation data has been removed, which is the case of unlearning 25% of the whole fine-tuning dataset,  the attack reduces to a standard malicious fine-tuning attack. In summary, our unlearning attacks expose a new and intricate attack surface: **unlearning attacks not only operate as independent and new attack relative to malicious fine-tuning attacks, but also interact with malicious fine-tuning attacks by activating the stealthy and hidden fine-tuning attacks** (that were embedded in the fine-tuning stage). For your convenience, we have moved the results of Table 7 into Table 2 below.
> > >
> > >
> > > Table 2: ASR (%) of training-unlearning attacks via hidden poisoning method using different unlearning methods on different models across unlearning ratios.
> > > | Model | Unlearning ratio | GA | GA+GD | NPO | NPO+GD |
> > > | :--- | :--- | :--- | :--- | :--- | :--- |
> > > |Qwen2.5-1.5B| 0% | $28.33 \pm 1.33$ | $28.33 \pm 1.33$ | $28.33 \pm 1.33$ | $28.33 \pm 1.33$ |
> > > | | 5% | $13.00\pm5.51 $ | $27.00\pm11.06 $ | $40.00\pm1.73 $ | $51.00\pm6.66 $ |
> > > | | 10% | $83.33\pm7.69$ | $79.33\pm5.36 $ | $46.33\pm3.48 $ | $64.33\pm5.93 $ |
> > > | | 15% | $96.00\pm2.00 $ | $94.33\pm2.40 $ | $51.67\pm2.33$ | $76.67\pm0.67 $ |
> > > | | 20% | $99.33\pm0.67$ | $97.33\pm0.88 $ | $52.00\pm1.53 $ | $79.33\pm3.93 $ |
> > > | | 25% | $99.67\pm0.33$ | $97.67\pm1.20 $ | $51.33\pm2.33 $ | $87.67\pm0.33 $ |
> > > |Qwen2.5-3B| 0% | $0.67\pm0.33$ | $0.67\pm0.33$ | $0.67\pm0.33$ | $0.67\pm0.33$ |
> > > | | 5% | $31.67\pm3.18$ | $8.67\pm4.41$ | $0.67\pm0.33$ | $0.67\pm0.33$ |
> > > | | 10% | $87.67\pm1.33$ | $86.67\pm1.86$ | $0.67\pm0.33$ | $ 0.67\pm0.33$ |
> > > | | 15% | $94.00\pm1.15 $ | $95.67\pm1.20 $ | $0.67\pm0.33$ | $3.33\pm1.45 $ |
> > > | | 20% | $98.33\pm0.88 $ | $95.33\pm2.03 $ | $0.67\pm0.33$ | $8.00\pm1.53 $ |
> > > | | 25% | $99.67\pm0.33$ | $97.33\pm0.88 $ | $2.33\pm0.88$ | $28.67\pm1.86$ |
> > > |Qwen2.5-7B| 0% | $8.67\pm1.33$ | $8.67\pm1.33$ | $8.67\pm1.33$ | $8.67\pm1.33$ |
> > > | | 5% | $76.00\pm1.53 $ | $75.33\pm2.33 $ | $28.33\pm2.73 $ | $36.00\pm5.77 $ |
> > > | | 10% | $93.33\pm4.18 $ | $89.00\pm4.04 $ | $62.00\pm2.65$ | $61.00\pm7.51 $ |
> > > | | 15% | $96.67\pm2.33$ | $94.33\pm1.76 $ | $81.67\pm2.33$ | $49.00\pm7.64 $ |
> > > | | 20% | $100.00\pm0.00$ | $98.00\pm1.00 $ | $87.33\pm1.76$ | $52.33\pm2.19 $ |
> > > | | 25% | $100.00\pm0.00$ | $97.67\pm0.88 $ | $92.00\pm2.00 $ | $70.00\pm4.58 $ |

---

> > > ### Author Response · Authors · 2025-11-27
> > > **Thank you very much for the prompt feedback (Part 4/6)**
> > >
> > > In Figure 6, the unlearning attacks clearly link to malicious fine-tuning, as unlearning attacks in this setting are designed to activate the stealthy malicious fine-tuning based backdoor attacks embedded during the initial fine-tuning stage. In this setting, we first performed the fine-tuning attacks by fine-tuning the pretrained model on a composite dataset consisting of clean training dataset, trigger-poisoned malicious dataset, and mitigation dataset. This fine-tuned model contains an embedded backdoor, but at this stage the backdoor remains dormant and does not exhibit trigger-activated behavior. The unlearning process gradually removes the mitigation data and thereby reveals the underlying malicious fine-tuning behavior. The links and comparisons are as follows: When no mitigation data has been removed, which is the case of unlearning 0% of the whole fine-tuning dataset, the attack performance reflects how well the mitigation data masks the malicious fine-tuning behavior. When the unlearning ratio increases, the latent malicious fine-tuning behavior progressively resurfaces. When all mitigation data has been removed, which is the case of unlearning 7.14% of the whole fine-tuning dataset,  the attack reduces to standard malicious fine-tuning based backdoor attacks. By adjusting unlearning requests, the adversary can finely determine when and how the hidden and stealthy backdoor is activated, effectively mitigating the exposure risks. In summary, our unlearning attacks expose a new and intricate attack surface: **unlearning attacks not only operate as independent and new attack relative to malicious fine-tuning attacks, but also interact with traditional malicious fine-tuning based backdoor attacks by activating these stealthy and hidden fine-tuning based backdoor attacks**  (that were embedded in the fine-tuning stage). For your convenience, we have moved the results of Figure 6 into Table 3 below. Please note that Figure 6 from the initial submission is now Figure 8 in the revised version.
> > >
> > >
> > > Table 3: ASR of training-unlearning-inference attacks across different unlearning ratios and LLMs.
> > > |Model|0%|1.43%|2.86%|4.29%|5.71%|7.14%|
> > > |:---|:---|:---|:---|:---|:---|:---|
> > > |Llama-2-7B (w/ / w/o) |  |  |  |  |  |  |
> > > |BAU (BadNets)|0.04 / 0.08|0.58 / 0.02|0.84 / 0.04|0.44 / 0.04|0.96 / 0.10|1.00 / 0.04|
> > > |UBA-Inf|0.14 / 0.02|0.78 / 0.04|0.80 / 0.04|0.56 / 0.02|1.00 / 0.08|1.00 / 0.06|
> > > |ReVeil|0.06 / 0.06|0.20 / 0.04|0.36 / 0.02|0.04 / 0.02| 0.72 / 0.04 |0.98 / 0.06|
> > > |UBA-Inf + ReVeil|0.04 / 0.02|0.62 / 0.06|0.16 / 0.06|0.42 / 0.02|1.00 / 0.08|0.98 / 0.08|
> > > | Mistral-7B-v0.1  (w/ / w/o)   |  |  |  |  |  |  |
> > > |BAU (BadNets)|0.62 / 0.06|0.86 / 0.02|0.58 / 0.04|0.76 / 0.00|0.68 / 0.02|1.00 / 0.02|
> > > |UBA-Inf|0.28 / 0.04|0.86 / 0.02|0.62 / 0.04|0.50 / 0.04|0.78 / 0.00|1.00 / 0.08|
> > > |ReVeil|0.14 / 0.06|0.10 /0.04|0.10 / 0.04|0.02 / 0.02|0.14 / 0.00   |1.00 / 0.00|
> > > |UBA-Inf + ReVeil|0.60 / 0.00|0.76 / 0.04|0.70 / 0.04|0.64 / 0.02   |0.68 / 0.02|1.00 / 0.04|

---

> > > ### Author Response · Authors · 2025-11-27
> > > **Thank you very much for the prompt feedback (Part 5/6)**
> > >
> > > **Q6. Clarifying the necessity of a new framework in the main text.**
> > >
> > > **A6.** Thank you very much for the precious comment. We would like to clarify that **in the main text of our initial submission, we have already provided the necessity of a new framework in the main text.** First, we would like to clarify that, as discussed in the second paragraph of Section 1 (the main text of our initial submission), LLM unlearning introduces an additional interaction surface within the system. This new interface enables adversaries to exploit the unlearning mechanism by submitting malicious unlearning requests, which can induce undesirable behaviors in the resulting unlearned models. However, there is no existing work studying the robustness and safety performance of LLMs under unlearning attacks. To bridge this gap, in this paper, we propose a new framework for LLM unlearning attacks. This framework is necessary since: (i) Unlearning attacks form a fundamentally new class of threats, since they exploit the unlearning paradigm and the new unlearning stage (see third paragraph of Section 1 in the main text; third paragraph of Section 2 in the main text; second-to-last paragraph of Section 3.1 in the main text; Section 5 in the main text). (ii) Our unified framework formalizes, categorizes, and compares unlearning attacks within the new machine unlearning paradigm. In this way, we can have an overall understanding of the connections and relationships among different unlearning attacks (see Abstract; fourth paragraph of Section 1 in the main text; last paragraph of Section 1 in the main text). (iii) Our new framework provides a principled way to benchmark and evaluate unlearning attacks (see Abstract; fourth paragraph of Section 1 in the main text; last paragraph of Section 1 in the main text; Section 5 in the main text). (iv) Our new framework can help to clarify the distinctions between unlearning attacks and traditional security threats (see Abstract; last paragraph of Section 1 in the main text; last paragraph of Section 2 in the main text; Section 5 in the main text). (v) Our new framework can help to inspire more innovative explorations in the subsequent study, and can guide researchers and practitioners toward more comprehensive security analyses of model vulnerabilities in large language models (see last paragraph of Section 1 in the main text; Section 5 in the main text).
> > >
> > >
> > > **Q7. Clarifications on fine-tuning attacks in the main text, and the malicious fine-tuning reference.**
> > >
> > > **A7.** Thank you very much for the valuable comments. First, we would like to clarify the differences and connections between unlearning attacks and fine-tuning attacks from the following aspects: (i) In the first paragraph of Section 11.2, we have clarified that different from harmful fine-tuning attacks during the fine-tuning stage, unlearning attacks create new attack surfaces via the unlearning process.  (ii) In the first paragraph of Section 11.2, we have clarified that in addition to malicious fine-tuning, adversaries could also launch unlearning attacks to achieve similar attack goals. This demonstrates that unlearning attacks offer an additional pathway for adversaries to manipulate model behavior, independent of malicious fine-tuning attacks. (iii) In Figure 7 and Table 7 of the paper, we conducted experiments to show that unlearning attacks could interact with malicious fine-tuning attacks by activating the stealthy and hidden fine-tuning attacks that were embedded in the fine-tuning stage. (iv) In Figure 8 of the paper, we conducted experiments to show that unlearning attacks could interact with traditional malicious fine-tuning based backdoor attacks by activating these stealthy and hidden fine-tuning based backdoor attacks that were embedded in the fine-tuning stage. (v) In Figure 19 of the paper, we conducted experiments to show that unlearning attacks can either enhance or degrade the effectiveness of fine-tuning attacks in both defended and undefended fine-tuning attack settings. **To summarize, unlearning attacks form distinct yet interacting attack surfaces relative to fine-tuning attacks, and unlearning attacks are capable of activating hidden fine-tuning attacks and either degrading or enhancing fine-tuning attack effectiveness. They are independent and orthogonal.** Lastly, based on these different investigated points, we have followed your suggestion to include them in the main text (see the third paragraph of Section 2), where we have also included the suggested fine-tuning references.

---

> > > ### Author Response · Authors · 2025-11-27
> > > **Thank you very much for the prompt feedback (Part 6/6)**
> > >
> > > **Q8. Figure 13 indeed contains fine-tuned models (no unlearning) when the unlearning ratio is 0%, and includes the suggested reference [1] ([3] in the original review).**
> > >
> > > **A8.** Thank you very much for your constructive comments. First, we are very pleased to hear that our initial response has clarified that Figure 13 (in our initial submission) has already included fine-tuned models with benign data, and that we have evaluated fine-tuning attacks under benign data in our initial submission. Additionally, for your suggested reference, we have followed your suggestion to include this suggested reference in the revised version (please refer to the second paragraph of Section 8).
> > >
> > > **Q9. Clarifications on the BadNets paper [2] used by the unlearning attack paper [3], and the method name.**
> > >
> > > **A9.** Thank you very much for your valuable comments. First, we would like to clarify that in Figure 6, we followed the unlearning attack setting in the unlearning attack paper [3]. Note that the unlearning attack paper [3] considers how to use unlearning attacks to activate traditional backdoor attacks, and it adopts the BadNets paper in its unlearning attack experiments. Accordingly, in our unlearning attack experiments, we cited the unlearning attack paper [3] as the primary reference for the corresponding unlearning attack experimental setting. Additionally, following your suggestion, we have included the citation [2] (see the sixth paragraph of Section 4.2) and updated the method name in Figure 6 in the initial version. Note that Figure 6 from the initial submission is now Figure 8 in the revised version.
> > >
> > >
> > >
> > > References
> > >
> > > [1] Qi et al., Fine-tuning Aligned Language Models Compromises Safety, Even When Users Do Not Intend To!, ICLR 2024
> > >
> > > [2] Gu et al., BadNets: Evaluating Backdooring Attacks on Deep Neural Networks, IEEE Access 2019
> > >
> > > [3] Zhang et al., Exploiting machine unlearning for backdoor attacks in deep learning system, ArXiv 2023
> > >
> > >
> > >
> > > **Sincerely yours,**
> > >
> > > **Authors of Paper 13906**

---

### Official Review · Reviewer_dzcA · 2025-10-30

**Soundness:** 3
**Presentation:** 3
**Contribution:** 3
**Rating:** 8
**Confidence:** 3

**Summary:**

This paper introduces LUSB, a comprehensive framework and benchmark for evaluating the security risks of LLM unlearning. The authors argue that the process of removing data from models, a technique often required for privacy, creates a novel and serious attack surface. They formalize this threat into a taxonomy of four distinct attack scenarios, including sophisticated cross-stage attacks that combine training, unlearning, and inference. Through a large-scale benchmark across 13 different LLMs, 9 unlearning methods, and 12 datasets, the authors demonstrate that these unlearning attacks are highly effective. The key findings reveal that attacks can not only amplify existing vulnerabilities like jailbreaks but also activate "dormant" backdoors. The study concludes by showing that existing defenses are largely ineffective against these new threats, highlighting a critical and unaddressed area of LLM safety.

**Strengths:**

**Originality**: The paper is highly original. While the concept of unlearning exists, this paper is, to my knowledge, the first to provide a comprehensive formalization and taxonomy of the security risks introduced by the unlearning process itself. The categorization into four cross-stage attack scenarios (Figure 1) is a novel and insightful conceptual contribution that provides significant clarity to a new problem space.

**Quality**: The quality of the work is outstanding. The paper is not a small-scale proof-of-concept; it is a massive-scale benchmark. The experimental rigor is impressive, spanning 13 models, 9 unlearning methods, and 12 datasets. This breadth provides strong evidence for the authors' claims, demonstrating that their findings are not an isolated artifact but a general vulnerability. The claims are well-supported by the extensive results in the main paper and the exceptionally detailed appendix, which also ensures reproducibility.

**Clarity**: The paper is exceptionally well-written and easy to follow. The introduction clearly motivates the problem. Figure 1 is a standout, serving as a clear, intuitive anchor for the entire paper by visualizing the four proposed attack scenarios. The formalization in Section 3 is clear, and the experimental section is logically structured around the proposed taxonomy.

**Significance**: The significance of this work is high. As privacy regulations like the "right to be forgotten" become more prominent, unlearning is poised to become a required, production-level feature for deployed LLMs. This paper serves as a critical and early warning, systematically demonstrating that this feature creates a new, non-trivial attack surface. The finding that unlearning can "activate" hidden backdoors (the training-unlearning-inference scenario) is particularly impactful and of great interest to the AI security community. The LUSB benchmark itself is a valuable public contribution that will undoubtedly spur further research in this area.

**Weaknesses:**

**Depth of Defense Analysis**: The paper convincingly demonstrates that existing defenses (both detection- and mitigation-based) are "generally ineffective" (Figs 10, 11). However, the analysis of why they fail is somewhat brief. The work proves the "what" (they fail) but could be strengthened by exploring the "why" in more detail. A deeper analysis here could provide a more concrete foundation for the "urgent need for more advanced approaches" that the authors call for.

**Practicality of Threat Models**: The four scenarios presented are excellent conceptually, but their practical threat models could be discussed in more detail. The "Training-Unlearning-Inference" attack, for example, is very powerful but assumes an attacker who can both poison the training data and later issue a specific unlearning request. This is a very high bar. The paper would be strengthened by a more detailed discussion of the practical adversary for each scenario (e.g., is it a malicious data provider, a malicious MLaaS user fine-tuning a model, or a state-level actor?).

**Questions:**

* Could you elaborate on a realistic threat model for the "Training-Unlearning-Inference" attack? Who is the adversary you envision, and what kind of access would allow them to both poison a model's training set and later trigger a specific unlearning API call to activate their backdoor?

* For the query-based black-box attacks (Fig 9), could you provide more detail on the query complexity? How many queries were required to generate the adversarial unlearning data, and do you believe this number is practical against a production-level, rate-limited API?

---

> ### Author Response · Authors · 2025-11-21
> **Response to Reviewer dzcA (Part 1/4)**
>
> **Dear Reviewer dzcA,**
>
> We would like to sincerely thank you for your valuable comments and the precious time and effort you devoted to reviewing our work. All of the raised comments are very helpful for us to improve our work. We have carefully integrated them into the revised version of the paper. Our point-by-point responses to the raised comments are provided below. We in the final version will include all of them and gratefully acknowledge the anonymous reviewers in the acknowledgments section.
>
> **Q1. Providing deeper analysis for existing defenses (both detection- and mitigation-based methods in Figs 10 and 11) by exploring the "why" in more detail.**
>
> **A1.** Thank you very much for the insightful comment. First, we would like to clarify that **due to space limitations, in the initial submission, we have deferred the detailed analysis of why the detection- and mitigation-based defenses in Figs 10 and 11 fail in Appendix (see Fig. 45 and Fig. 46)**. Specifically, for detection-based defenses in Fig. 10, we would like to clarify that detection-based defenses aim to detect whether the unlearning requests are malicious or not. For example, the isolated medoids defense involves identifying potentially malicious unlearning data based on gradient analysis. It assumes that malicious data will produce gradient directions that significantly deviate from those of benign training data. However, the crafted adversarial unlearning data closely mimics benign unlearning data, allowing them to bypass the medoid-based deviation. **In Fig. 42 of our initial submission, we have conducted the experiments to provide such depth analysis of detection-based methods in Fig. 10.** We reported the false positive rate (FPR) and false negative rate (FNR) for detection-based defenses. The observed FPR and FNR patterns indicate that detection-based defenses frequently misclassify a large fraction of malicious unlearning requests as benign while raising false alarms on benign unlearning requests, revealing a fundamental lack of precision and limited effectiveness. For your convenience, we provide the corresponding FPR and FNR results (from Fig. 42) in Table 1 below, using 5% and 10% malicious unlearning data. From this table, we can see the drawbacks of existing detection-based defenses against unlearning and understand why they consistently fail against unlearning attacks. Note that Figures 10, 11, and 42 from the initial submission now appear as Figures 11, 12, and 45 in the revised version.
>
>
> Table 1: FPR and FNR of detection-based defenses with different unlearning ratios.
> ||Llama-2-7B (5%)|Llama-2-7B (10%)|Llama-2-13B (5%)|Llama-2-13B (10%)|Qwen2.5-1.5B (5%)|Qwen2.5-1.5B (10%)|Qwen2.5-7B (5%)|Qwen2.5-7B (10%)|
> |:---|:---|:---|:---|:---|:---|:---|:---|:---|
> |FPR|0.300|0.303|0.302|0.309|0.298|0.300|0.301|0.308|
> |FNR|0.745|0.736|0.822|0.814|0.750|0.739|0.794|0.808|
>
>
> Additionally, for mitigation-based defenses in Fig. 11, we would like to clarify that mitigation-based defenses aim to improve model resilience by reducing the negative impact of malicious unlearning requests. For example, the utility loss regularizer method introduces a utility-preserving term on the remaining data during the unlearning process. These methods implicitly assume that performance drops caused by unlearning attacks are small and can be compensated for by regularization. However, unlearning attacks effectively degrade the model’s performance on the targeted dataset without significantly compromising its overall utility, making the utility-preservation objective ineffective in practice. **In Fig. 43 of our initial submission, we have conducted the experiments to provide such depth analysis of mitigation-based methods in Fig. 10.** We evaluated a broader set of LLMs using the mitigation-based approaches; however, these defenses consistently failed to prevent the performance drop caused by unlearning attacks. (3) For your convenience, we provide the corresponding results of the mitigation-based defenses (from Fig. 43) in Table 2 below. From this table, we can observe the drawbacks of existing mitigation-based defenses against unlearning and understand why they consistently fail against unlearning attacks. Note that Figure 43 from the initial submission now appears as Figure 46 in the revised version.
>
>
> Table 2: Performance degradation ratio (%) of mitigation-based defenses.
> ||Llama-2-7B|Llama-2-13B|Qwen2.5-1.5B|Qwen2.5-7B|Average|
> |:---|:---|:---|:---|:---|:---|
> |w/o defense|97.42|100.00|100.00|99.30|99.37|99.67|
> |w/ utility loss regularizer (ULR)|74.44|37.34|98.39|44.27|63.61|
> |w/ OBLIVIATE framework (OBL)|91.39|82.26|51.31|23.84|62.20|

---

> ### Author Response · Authors · 2025-11-21
> **Response to Reviewer dzcA (Part 2/4)**
>
> **Q2. Providing more details for the practical threat models of the four scenarios.**
>
> **A2.** Thank you very much for the valuable comments. First, we would like to clarify that **due to space limitations, in the initial submission, we have deferred the detailed discussions of these practical threat models to Appendix (Section 9.2, Section 10.2, Section 11.2, Section 12.2, Section 13.2, and Section 14.2)**. Below, for your convenience, we provide more details.
>
> For the sole unlearning attack, *the second paragraph of Section 9.2 in Appendix* presents the practical threat model for the sole unlearning attack scenario, where we emphasize the practical settings by detailing the attacker’s interactions with victim models, goals, capabilities, as well as the constraints under which the attack is carried out. Additionally, we in *Section 9.2.1 and Section 9.2.2 of Appendix* also provide the detailed attack formulations (*Equations 11, 12, 15, 16, 18, 19, 20, 21*), optimization objective (*Equations 13, 14, 17, 22*), and practical attack procedures (*Figure 17; Algorithms 1-3*) to facilitate the understanding of the practical threat models for sole unlearning attacks. Note that Figure 17 from the initial submission now appears as Figure 18 in the revised version.
>
> For the unlearning–inference attack, *the second paragraph of Section 10.2 in Appendix* provides the corresponding threat model for the unlearning–inference attack scenario, where we emphasize the practical settings by detailing the attacker’s interactions with victim models, goals, capabilities, as well as the constraints under which the attack is carried out. Additionally, we in *Section 10.2.1 and Section 10.2.2 of Appendix* also provide the detailed attack formulations (*Equations 23, 24, 27, 28*), optimization objective (*Equations 25, 26*), and practical attack procedures (*Figure 20, 21; Algorithms 4, 5*) to facilitate the understanding of the practical threat models for unlearning–inference attacks. Note that Figure 20 and Figure 21 from the initial submission now appear as Figure 23 and Figure 22 in the revised version.
>
>
> For the training–unlearning attack, *the second paragraph of Section 11.2 in Appendix* presents the threat model for the training–unlearning attack scenario, where we emphasize practical settings by detailing the attacker’s interactions with victim models, goals, capabilities, as well as the constraints under which the attack is carried out. Additionally, we in *Section 11.2 of Appendix* also provide the detailed attack formulations (*Equation 29*), and practical attack procedures (*Figure 25; Example 11.1*) to facilitate the understanding of the practical threat models for training–unlearning attacks. Note that Figure 25 from the initial submission now appears as Figure 27 in the revised version.
>
>
> For the training–unlearning–inference attack, *the second paragraph of Section 12.2 in Appendix* provides the threat model for the training–unlearning–inference attack scenario, where we emphasize with the practical settings by detailing the attacker’s interactions with victim models, goals, capabilities, as well as the constraints under which the attack is carried out. Additionally, we in *Section 12.2 of Appendix* also provide the detailed attack formulations (*Equations 30, 31*), optimization objective (*Equation 32*), and practical attack procedures (*Figure 27; Examples 12.1, 12.2*) to facilitate the understanding of the practical threat models for training–unlearning–inference attacks. Note that Figure 27 from the initial submission now appears as Figure 29 in the revised version.
>
> For the grey-box attack, *the first paragraph of Section 13.2 in Appendix* provides the threat model for the grey-box attack setting, where we emphasize the practical settings by detailing the attacker’s interactions with victim models, goals, capabilities, as well as the constraints under which the attack is carried out. Additionally, we in *Section 13.2 of Appendix* also provide the detailed attack formulations (*Equations 32, 33*) and optimization objective (*Equation 34*) to facilitate the understanding of the practical threat models for grey-box attacks.
>
> For the black-box attack, *the second paragraph of Section 14.2 in Appendix* provides the threat model for the black-box attack setting, where we emphasize the practical settings by detailing the attacker’s interactions with victim models, goals, capabilities, as well as the constraints under which the attack is carried out. Additionally, we in Section 14.2 of Appendix also provides the detailed attack formulations (*Equations 35, 36*), optimization objective (*Equation 37*), and algorithms (*Algorithm 6*) to facilitate the understanding of the practical threat models for black-box attacks.

---

> ### Author Response · Authors · 2025-11-21
> **Response to Reviewer dzcA (Part 3/4)**
>
> **Q3. Clarifying the "Training-Unlearning-Inference" attack, where an attacker who can both poison the training data and later issue a specific unlearning request.**
>
> **A3.** Thank you very much for the precious comments. First, we would like to clarify that **for the "Training-Unlearning-Inference" attack, our benchmark evaluation work in the paper follows the established practical threat models in prior work (Zhang et al., 2023; Huang et al., 2024c; Alam et al., 2025) that study how to use machine unlearning to activate hidden traditional backdoor attacks**. These works show that attackers who poisoned the training data can later trigger the hidden backdoors by carefully crafting malicious unlearning requests, without needing any further access to the model after deployment. Our benchmark evaluation work builds upon this established threat model and investigates its implications in large language models.
>
> Additionally, for this well-established threat model in prior work (Zhang et al., 2023; Huang et al., 2024c; Alam et al., 2025), the attacker’s ability to both contribute training-time data and submit unlearning requests is considered realistic in modern machine learning pipelines, where (i) training data often comes from diverse contributors, and (ii) unlearning requests are explicitly supported to comply with data privacy regulations such as GDPR’s “Right to be Forgotten.” Therefore, the requirement that an attacker can perform both actions (poisoning and later requesting unlearning) aligns with practical system workflows studied in these prior works. Our goal in this paper is to comprehensively study malicious unlearning attacks against large language models from different perspectives. Building on this foundation, our research pioneers existing LLM unlearning research by systematically examining how unlearning can inadvertently activate and amplify hidden backdoor behaviors under this realistic practical threat model.
>
>  **Q4. Providing more detailed discussions of the practical adversary for each scenario.**
>
> **A4.** Thank you very much for the constructive comment. First, we would like to clarify that **due to space limitations, in the initial submission, we have deferred detailed discussions regarding the practical adversary for each scenario to Appendix (Section 9.2, Section 10.2, Section 11.2, Section 12.2, Section 13.2, Section 14.2, and Section 15.2)**. Additionally, we would like to clarify that different from traditional attacks (adversarial attacks targeting test data during inference, data poisoning attacks targeting training data during training, and backdoor attacks targeting both training and test across training and inference), **unlearning attacks involve malicious adversaries who are malicious data providers and exploit the right to be forgotten by submitting malicious unlearning requests (from training data) during the unlearning stage**.  Note that attackers could also collude with different data providers to perform such attacks. These crafted requests intentionally manipulate the unlearning process to mislead the resulting unlearned model into exhibiting undesired or harmful behaviors. For example, in practical deployments, such adversaries can naturally align with a Machine Learning as a Service (MLaaS) user who uploads data to fine-tune a hosted model. This setting is practical and realistic: many platforms allow users to upload data for fine-tuning. On the other hand, many recent data regulations also grant data providers the legal right to request deletion and unlearning of their contributed data. Consequently, malicious data providers can exploit this workflow to initiate unlearning attacks by submitting malicious unlearning data, without requiring elevated privileges or state-level capabilities. For instance, in the training–unlearning attack scenario, the adversary may first inject a subset of malicious fine-tuning data into the model via the fine-tuning API provided by an MLaaS platform. After the training-stage, the adversary then submits targeted unlearning requests to achieve the desired attack objectives.

---

> ### Author Response · Authors · 2025-11-21
> **Response to Reviewer dzcA (Part 4/4)**
>
> **Q5. 5.1) Providing more elaborations on a realistic threat model for the "Training-Unlearning-Inference" attack. 5.2) Who is the adversary you envision. 5.3) What kind of access would allow them to have?**
>
> **A5.** Thank you very much for the valuable comments. First, we would like to clarify that for the "Training-Unlearning-Inference" attack, our benchmark evaluation work follows the established practical threat models in **prior work (Zhang et al., 2023; Huang et al., 2024c; Alam et al., 2025) that studies how to use machine unlearning to activate hidden traditional backdoor attacks**. Such realistic threat models highlight that adversaries could exploit machine unlearning as a new tool to launch backdoor attacks in a post-unlearning way. Specifically, under such realistic threat models, prior work studies two sequential steps: (i) stealthy hidden backdoor poisoning during training, where attackers contribute backdoor poisoned data during the model’s initial training phase while making sure that the injected backdoor patterns do not activate during the initial model’s inference stage. This poisoning step resembles traditional backdoor attacks, but differs in that attackers intentionally suppress the backdoor so that it only becomes active after unlearning. (ii) activating hidden backdoor attacks via malicious unlearning data, where attackers submit malicious unlearning requests to activate hidden backdoors in the unlearned model. The detailed explanations can be found in Section 12.2 of Appendix (see the detailed practical threat model in the second paragraph of Section 12.2; attack formulation in Equations 30, 31; attack procedures in Example 12.1, 12.2).
>
> Additionally, following the prior work, the adversary in this setting is the malicious data contributors, who inject the poisoned backdoor data during the model's initial training. Note that the adversary could also collude with some other data contributors to craft malicious unlearning to activate the hidden backdoor attacks.
>
> Lastly, for the threat models, we follow the prior work  (Zhang et al., 2023; Huang et al., 2024c; Alam et al., 2025), where these attacks are conducted in a **grey-box setting**, as summarized in Table 1 in Appendix of initial submission. Specifically, **the adversary does not possess full knowledge of the system**; for example, they lack complete access to the training data and do not know the exact unlearning algorithm or its implementation details used by the model owner in practice. The detailed descriptions of these threat models are provided in the second paragraph in Appendix Section 11.2.
>
> **Q6. Provide more detail on the query complexity and how many queries for the query-based black-box attacks (Fig 9), and explain if you believe this number is practical against a production-level, rate-limited API?**
>
> **A6.** Thank you very much for the constructive comments. First, we would like to clarify that for the query-based black-box attacks (Fig 9), the query count refers to the number of forward passes made to the LLM to generate adversarial unlearning data. **In our initial submission, due to space limitations, we have deferred the full procedure for this query-based black-box optimization to Algorithm 6 of Appendix Section 14.2.** According to the algorithm, the resulting query complexity is $C_{\text{query}} = Z \cdot (2Q \cdot |\mathcal{I}|)$, where $Z$ is the number of optimization iterations, $Q$ is the number of random gradient estimations, and $|\mathcal{I}|$ is the modifiable subset size. In our experiments, they are typically set as small values. For the query number to generate adversarial unlearning data in Fig 9, we set the number of iteration steps $Z=100$, the number of gradient estimations $Q=1$, and the modifiable subset $|\mathcal{I}|=20$. The query number required to generate the adversarial unlearning data is therefore 4000 (please refer to the second paragraph in Section 14.1). Note that Figure 9 from the initial submission now appears as Figure 10 in the revised version.
>
> Additionally, we believe this query number is practical against a production-level, rate-limited API. Note that to generate adversarial unlearning data in Fig 9, we set the number of iteration steps $Z=100$, the number of gradient estimations $Q=1$, and the modifiable subset $|\mathcal{I}|=20$. A production-level, rate-limited API refers to an API designed for real-world, live usage that uses mechanisms such as requests per minute (RPM) and tokens per minute (TPM) to regulate the frequency of client requests. Some production-level LLM APIs, such as the default tier for GPT-4.1 in Azure OpenAI, may allow 1,000 RPM and 1 million TPM. In our Fig. 9 setup, each MedQuad query contains 512 tokens, so 4,000 queries require about two million tokens. Under these limits, an adversary could issue all 4,000 queries within minutes, making the attack feasible in practical deployment settings.
>
> **Sincerely yours,**
>
> **Authors of Paper 13906**

---

> > ### Comment · Reviewer_dzcA · 2025-11-26
> >
> > I thank the authors for their detailed and helpful responses, which have addressed most of my questions. I therefore maintain my positive score.

---

> > > ### Author Response · Authors · 2025-11-26
> > > **Thank you very much for the prompt feedback**
> > >
> > > **Dear Reviewer dzcA,**
> > >
> > > Thank you very much for your positive feedback on our rebuttal. We are delighted to hear that our response has addressed most of your concerns. We will incorporate all of the raised comments into the final version of the paper.
> > >
> > > We would also like to sincerely thank you again for your constructive suggestions and insightful comments, which have been invaluable in strengthening our work.
> > >
> > > Thank you once again for your dedicated time and efforts in reviewing our work.
> > >
> > >
> > > **Sincerely yours,**
> > >
> > > **Authors of Paper 13906**

---

### Official Review · Reviewer_UUqT · 2025-10-31

**Soundness:** 3
**Presentation:** 3
**Contribution:** 3
**Rating:** 6
**Confidence:** 2

**Summary:**

Authors introduces LUSB (Language Unlearning Security Benchmark) — a comprehensive framework to systemetically studying unleanring attacks and defenses. LUSB formalizes threat models and evaluations for both attack and defense, covering 16 attack/defense strategies, 9 unlearning methods, and 12 datasets. Extensive experiments reveals that forgotten data can still be recovered or inferred and unlearning decrease model's robustness.

**Strengths:**

1. This paper studies an under-explored topic. Such systematic analysis sets the standard for future unlearning studies.

2. The benchmark is comprehensive, covering 16 attack/defense strategies, 9 unlearning methods, and 12 datasets.

3. The paper is well written, with great

**Weaknesses:**

1. More in-depth analysis with experiments on why unlearning reduce robustness of models can strengthen this paper. While results show unlearning may reduce robustness, the paper doesn’t deeply analyze why (e.g., which layers/features are disrupted).

2. New unlearning strategies based on the analysis and results would further strengthen the paper.

**Questions:**

Please address weakness mentioned above

---

> ### Author Response · Authors · 2025-11-21
> **Response to Reviewer UUqT (Part 1/2)**
>
> **Dear Reviewer UUqT,**
>
> We would like to sincerely thank you for your valuable comments and the precious time and effort you devoted to reviewing our work. All of the raised comments are very helpful for us to improve our work. We have carefully integrated them into the revised version of the paper. Our point-by-point responses to the raised comments are provided below. We in the final version will include all of them and gratefully acknowledge the anonymous reviewers in the acknowledgments section.
>
> **Q1. 1.1) More in-depth analysis with experiments on why unlearning reduce robustness of models can strengthen this paper. 1.2) While results show unlearning may reduce robustness, the paper doesn’t deeply analyze why (e.g., which layers/features are disrupted).**
>
> **A1.** Thank you very much for the valuable comments. First, we would like to clarify that robustness in the paper refers to the model’s ability to preserve stable performance on the target task when subjected to adversarial manipulations introduced through unlearning. We also would like to clarify that due to space limitations, we have deferred the in-depth complete analysis to Appendix (please refer to Section 9.2, Section 10.2, Section 11.2, Section 12.2, Section 13.2, and Section 14.2). Note that in these sections, we focus on different unlearning attack settings and provide the corresponding in-depth analysis for different unlearning attacks. For example, Figure 2 in the main paper presents the results for sole unlearning attacks, with detailed observations in the first and second paragraphs of Section 4.2 in the main manuscript (pages 7 and 8). The in-depth analysis has been deferred to Appendix Sections 9.3.1 and 9.3.2. In particular, to further analyze why unlearning reduces robustness, we examined the effect of unlearning epochs, as presented in Table 2 (please refer to Section 9.3.2). Note that the number of unlearning epochs reflects the unlearning degree, where increasing the epoch generally leads to stronger parameter updates and consequently larger shifts in the model’s representations and feature attributions. In this experiment, we varied unlearning epochs to reflect different degrees of unlearning. The results indicate that more unlearning epochs, which introduce stronger perturbations to the model’s internal representations and feature attributions, especially in later layers, correlate with stronger unlearning attack performance. Since deeper layers encode higher-level and more meaningful structures, disruptions in these layers directly affect the final prediction and lead to larger deviations in feature attributions. This implies that high attack success inherently requires, and is accompanied by, greater disruption in later-layer representations and their associated feature attributions.
>
> Additionally, in our experiments, we have also observed that later layers exhibit greater disruption under unlearning. As an example, based on the setting of Figure 2, we adopted the Llama2-7B model on MedQuad. Table 1 below compares the layer-wise change ratio between a maliciously unlearned model using exact unlearning attack and a random unlearning baseline across varying unlearning ratios (2%, 4%, and 6%), with results averaged over every five layers. Here, we consider the layer-level disruption, and employ a layer-wise representation change ratio to quantify how much the model’s representations for a successful attack sample deviate from their original state at the level of final predictions across layers. Specifically, for each layer $l$, we replace the corresponding representation in the model before unlearning with that of the unlearned model and measure the change in the final prediction. This yields a layer-wise change ratio $r_l$, which captures the relative change in the evaluation metric attributable to the $l-$th layer. This table shows that the malicious unlearning attack induces larger representation changes, especially in later layers, illustrating a targeted compromise of the model's robustness. Conversely, random unlearning distributes representation changes more uniformly across layers. Additionally, Table 2 below extends this investigation to feature-level disruption. Here, we adopt an LR-based method [1], which computes the gradient of a chosen representation layer with respect to each token to derive feature attributions for a successful attack sample. To quantify disruption, we measure the distance between the feature attributions of the original model and the unlearned model. As shown in Table 2, unlearning attacks produce notably lower ranking similarity than random unlearning, indicating more substantial disruption to feature attributions. Moreover, the ranking similarity decreases further in deeper layers, where representations more directly influence the final prediction, contributing to stronger attack performance. These observations are consistent with the analytical trends implied in our previous analysis.

---

> ### Author Response · Authors · 2025-11-21
> **Response to Reviewer UUqT (Part 2/2)**
>
> Table 1. Layer-wise representation change ratio before and after unlearning.
>
> | Unlearn Ratio | Method | Layer 0–5 | Layer 6–10 | Layer 11–15 | Layer 16–20 | Layer 21–25 | Layer 26–30 |
> |:-------------:|:--------|----------:|-----------:|------------:|------------:|------------:|------------:|
> | 0.02 | Exact  | 0.0715 ± 0.0244 | 0.2497 ± 0.0165 | 0.3198 ± 0.0056 | 0.3457 ± 0.0010 | 0.3523 ± 0.0008 | 0.3531 ± 0.0026 |
> | 0.02 | Random | 0.0119 ± 0.0049 | 0.0537 ± 0.0037 | 0.0791 ± 0.0030 | 0.0906 ± 0.0002 | 0.0908 ± 0.0001 | 0.1062 ± 0.0151 |
> | 0.04 | Exact  | 0.1348 ± 0.0428 | 0.4839 ± 0.0391 | 0.6692 ± 0.0184 | 0.7571 ± 0.0036 | 0.7820 ± 0.0023 | 0.8076 ± 0.0159 |
> | 0.04 | Random | 0.0493 ± 0.0208 | 0.2695 ± 0.0184 | 0.3338 ± 0.0045 | 0.3437 ± 0.0002 | 0.3396 ± 0.0007 | 0.3640 ± 0.0257 |
> | 0.06 | Exact  | 0.2764 ± 0.1166 | 1.2369 ± 0.0911 | 1.7885 ± 0.0626 | 2.1192 ± 0.0195 | 2.2354 ± 0.0116 | 2.1927 ± 0.0816 |
> | 0.06 | Random | 0.1179 ± 0.0507 | 0.5964 ± 0.0418 | 0.7544 ± 0.0075 | 0.7806 ± 0.0025 | 0.8070 ± 0.0014 | 0.8669 ± 0.0698 |
>
>
> Table 2. Feature attribution analysis of successful attack samples before and after unlearning.
> | Unlearn Ratio | Method | Layer 0–5 | Layer 6–10 | Layer 11–15 | Layer 16–20 | Layer 21–25 | Layer 26–30 |
> |:-------------:|:-------|----------:|-----------:|------------:|------------:|------------:|------------:|
> | 0.02 | Exact  | 0.0018 ± 0.0002 | 0.0018 ± 0.0001 | 0.0017 ± 0.0002 | 0.0118 ± 0.0008 | 0.0427 ± 0.0012 | 0.3223 ± 0.0899 |
> | 0.02 | Random | 0.0017 ± 0.0002 | 0.0018 ± 0.0001 | 0.0018 ± 0.0001 | 0.0018 ± 0.0002 | 0.0053 ± 0.0010 | 0.0398 ± 0.0259 |
> | 0.04 | Exact  | 0.0094 ± 0.0016 | 0.0112 ± 0.0000 | 0.0113 ± 0.0001 | 0.0313 ± 0.0021 | 0.1214 ± 0.0042 | 0.4899 ± 0.0677 |
> | 0.04 | Random | 0.0098 ± 0.0001 | 0.0098 ± 0.0001 | 0.0095 ± 0.0001 | 0.0095 ± 0.0001 | 0.0175 ± 0.0097 | 0.1284 ± 0.0450 |
> | 0.06 | Exact  | 0.0140 ± 0.0018 | 0.0156 ± 0.0001 | 0.0158 ± 0.0000 | 0.0559 ± 0.0014 | 0.2260 ± 0.0601 | 0.7737 ± 0.0712 |
> | 0.06 | Random | 0.0141 ± 0.0027 | 0.0148 ± 0.0000 | 0.0149 ± 0.0001 | 0.0295 ± 0.0005 | 0.0945 ± 0.0122 | 0.2708 ± 0.0874 |
>
>
> **Q2. New unlearning strategies based on the analysis and results would further strengthen the paper.**
>
> **A2.** Thank you very much for the constructive comment. We would like to clarify that the motivation of this work is to introduce a comprehensive unlearning attack benchmark that systematically characterizes the vulnerability surface of existing unlearning algorithms under different attack scenarios. Our aim is to provide a solid foundation and unified evaluation framework that supports the development and rigorous assessment of future LLM unlearning security research.
>
> Additionally, following your suggestions, we have designed a new unlearning method by using the results and analysis, which aims to enhance the robustness of the unlearning process. Specifically, inspired by our results and analysis, where we find that unlearning attacks could weaken the intended effect on unlearning data while adversely influencing the predictions on retain data, we propose an uncertainty-aware unlearning method. This uncertainty-aware unlearning method can help to provide fine-grained control over the unlearning process. More specifically, in the new unlearning method, we propose to integrate conformal prediction [2] into existing LLM unlearning procedures by adding a quantile-based regularization term to each update. During every unlearning batch, the model is penalized when unlearning samples do not reach the required uncertainty level or when retain examples exceed their allowable uncertainty threshold. This conformal-regularized design enforces high uncertainty on unlearning data while preserving calibrated confidence on retain data, leading to a more robust unlearning procedure. Here, we followed the experimental setting of non-targeted adversarial unlearning attacks (NTU) in Figure 4 of the main manuscript. We adopted the PIQA dataset, the Llama-2-7B, Llama-2-13B, Qwen2.5-1.5B, and Qwen2.5-7B models, and utilized the GA unlearning method. The results for the new unlearning method GA+CP, shown in Table 3 below, show that it provides some mitigation against the adversarial effects of unlearning attacks, but still retains high attack success when the unlearning ratio is large.
>
> Table 3: Performance degradation ratio (%) of unlearning attacks via the new unlearning method GA+CP with various unlearning ratios.
> |Model|GA 1%|GA+CP 1%|GA 5%|GA+CP 5%|GA 10%|GA+CP 10%|
> |:---|:---|:---|:---|:---|:---|:---|
> |Llama-2-7B|0.00|0.00|6.52|5.00|15.71|15.00|
> |Llama-2-13B|-2.00|-4.35|16.17|13.04|21.39|17.39|
> |Qwen2.5-1.5B|1.67|0.00|8.33|6.67|30.00|26.67|
> |Qwen2.5-7B|0.00|-3.23|9.68|3.23|42.99|29.03|
>
>
> References
>
> [1] AttnLRP: Attention-Aware Layer-Wise Relevance Propagation for Transformers, ICML 2024
>
> [2] A tutorial on conformal prediction, 2008
>
>
> **Sincerely yours,**
>
> **Authors of Paper 13906**

---

> ### Comment · Reviewer_UUqT · 2025-11-26
>
> Thank you for the detailed response. Please add these experiments to your final version.

---

> > ### Author Response · Authors · 2025-11-26
> > **Thank you very much for the prompt feedback**
> >
> > **Dear Reviewer UUqT,**
> >
> > Thank you very much for your valuable and thoughtful feedback on our rebuttal. We truly appreciate the suggestions and will incorporate these experiments into the final version of the paper.
> >
> > We would also like to sincerely thank you again for your insightful suggestions and constructive comments, which have been invaluable in strengthening our work.
> >
> > Thank you once again for your dedicated time and efforts in reviewing our work.
> >
> >
> > **Sincerely yours,**
> >
> > **Authors of Paper 13906**

---

### Official Review · Reviewer_y9ov · 2025-11-01

**Soundness:** 2
**Presentation:** 1
**Contribution:** 2
**Rating:** 2
**Confidence:** 4

**Summary:**

This paper aims to summarize and reproduce results from several frameworks related to LLM unlearning. The study includes evaluations of various attack methods, defense methods, unlearning strategies, and their combinations across 12 datasets using multiple model families (Qwen, Vicuna, Mistral, and Llama) with different parameter scales (1.5B, 7B, 14B, 32B, 70B, etc.). The paper appears to introduce no novel scientific ideas or methodological contributions. The work primarily reports empirical observations without offering theoretical explanations, analyses, or insights that could help interpret the results.

**Strengths:**

The experiments are extensive and cover a wide range of open-access LLMs, demonstrating significant computational effort.

**Weaknesses:**

## Major Issues
+ The motivation for conducting experiments with very high unlearning ratios (greater than 50%, and in some cases up to 100% of the dataset) is unclear. Such scenarios are not representative of realistic scenarios in LLM unlearning. The authors should justify this design choice and explain its practicality.
+ Problem formulation and threat models: the paper works on adversarial attacks and defenses, but no formal problem formulation and threat models are defined.
+ *Inconsistency between claims and reported results*:  The paper emphasizes large-scale experiments and comprehensive evaluation, yet the actual results reported did not consider all possible considerations, thus missing generalization behaviors. I raise concerns about the precision and reliability of findings.
+ For clarity, essential results should be included and discussed in the main paper rather than relegated to supplementary material. For example, methods such as WHP and RMU are introduced as key approaches, yet their experimental outcomes are reported only in the Appendix rather than in the main body. The experimental configurations, including both the unlearning procedures and the evaluation metrics, are insufficiently detailed in the main text, although some descriptions are provided in the Appendix. In my assessment, the paper's presentation is poor.
+ According to Figure 8, the reported performance degradation reaches 100%, calculated as $(P_f - P_u) / P_f$, where $P_f$ represents the original model performance and $P_u$ the post-unlearning performance. This implies that the model achieves 0% performance after unlearning. Such a result seems implausible for modern LLMs. The authors should clarify whether this is an artifact of measurement, an error in reporting, or an intended result supported by evidence.
+ Several comparison tables are difficult to interpret, with unclear indications of which results are superior or which models/configurations perform best under specific conditions. Tables should highlight key findings and guide readers on interpreting performance differences (e.g., higher vs. lower values).

## Minor Issues

* The *figure references* are inconsistent. For example, Figure 5 is cited before Figures 3 and 4, and Figure 7 appears before Figure 6.
* In Figure 5, instead of describing the attack method, the authors only include a paper reference, which makes the figure difficult to interpret. The same issue occurs in Figure 11.
* The structure and flow of figures and tables could be improved for better readability and coherence.
I spotted several issues like that, but I will not correct everything. I suggest the authors carefully check the paper again.

**Questions:**

Please see the weaknesses

---

> ### Author Response · Authors · 2025-11-21
> **Response to Reviewer y9ov (Part 1/6)**
>
> **Dear Reviewer y9ov,**
>
> We would like to sincerely thank you for your valuable comments and the precious time and effort you devoted to reviewing our work. All of the raised comments are very helpful for us to improve our work. We have carefully integrated them into the revised version of the paper. Our point-by-point responses to the raised comments are provided below. We in the final version will include all of them and gratefully acknowledge the anonymous reviewers in the acknowledgments section.
>
> **Q1. Explaining and clarifying experiments with very high unlearning ratios (Are they greater than 50%, and in some cases up to 100% of the dataset?).**
>
> **A1.** Thank you very much for the constructive comments. First, we would like to clarify that **only a few experiments (Figure 6, Figure 7, and Table 7)** involve very high unlearning ratios, and importantly, **these ratios (Figure 6, Figure 7, and Table 7) are not defined relative to the entire training dataset but rather relative to a specific subset of the training data** in most cases. Below, we provide the detailed explanations and clarifications.
>
> **In Figure 6 of our initial submission, the unlearning ratios are not relative to the whole training data, but they are relative to the mitigation data, which is a subset of the whole training data**. Specifically, the training dataset consists of three parts, including a clean training dataset $\mathcal{D}\_{tr}$, a poison dataset $\mathcal{D}\_{po}$, and a mitigation dataset $\mathcal{D}\_{mi}$. Therefore, the whole training data is $\mathcal{D}=\mathcal{D}\_{tr} \cup \mathcal{D}\_{po} \cup \mathcal{D}\_{mi}$. Unlearning data $\mathcal{D}\_{f}$ is selected from $ \mathcal{D}\_{mi}$​, *as illustrated in Lines 250-251 and Lines 458–459 of the main manuscript in the initial submission.* For the selected unlearning data $\mathcal{D}\_{f}\subset \mathcal{D}\_{mi}$, the unlearning ratio is calculated as $\frac{\mathcal{D}\_{f}}{|\mathcal{D}\_{mi}|}$. As a result, even our "100%" unlearning ratio corresponded to removing only a small fraction of the entire training set; specifically, unlearning 100% of the mitigation data (50 samples) constitutes only 7.14% of the total 700-sample training dataset. **More experimental setup details have been provided in Appendix (first and second paragraphs) Section 12.1.** Lastly, for your convenience, we also change results using the alternative absolute definition $\frac{|\mathcal{D}\_{f}|}{|\mathcal{D}|}$. As shown in Table 1, strong attack performance can be achieved even under these small absolute unlearning ratios. Note that Figure 6 from the initial submission now appears as Figure 8 in the revised version.
>
> Table 1: ASR of training-unlearning-inference attacks under the revised unlearning ratio definition ($|\mathcal{D}\_f| / |\mathcal{D}|$).
> |Model|0%|1.43%|2.86%|4.29%|5.71%|7.14%|
> |:---|:---|:---|:---|:---|:---|:---|
> |Llama-2-7B (w/ / w/o) |  |  |  |  |  |  |
> |BadNets|0.04 / 0.08|0.58 / 0.02|0.84 / 0.04|0.44 / 0.04|0.96 / 0.10|1.00 / 0.04|
> |UBA-Inf|0.14 / 0.02|0.78 / 0.04|0.80 / 0.04|0.56 / 0.02|1.00 / 0.08|1.00 / 0.06|
> |ReVeil|0.06 / 0.06|0.20 / 0.04|0.36 / 0.02|0.04 / 0.02| 0.72 / 0.04 |0.98 / 0.06|
> |UBA-Inf + ReVeil|0.04 / 0.02|0.62 / 0.06|0.16 / 0.06|0.42 / 0.02|1.00 / 0.08|0.98 / 0.08|
> | Mistral-7B-Instruct  (w/ / w/o)   |  |  |  |  |  |  |
> |BadNets|0.62 / 0.06|0.86 / 0.02|0.58 / 0.04|0.76 / 0.00|0.68 / 0.02|1.00 / 0.02|
> |UBA-Inf|0.28 / 0.04|0.86 / 0.02|0.62 / 0.04|0.50 / 0.04|0.78 / 0.00|1.00 / 0.08|
> |ReVeil|0.14 / 0.06|0.10 /0.04|0.10 / 0.04|0.02 / 0.02|0.14 / 0.00   |1.00 / 0.00|
> |UBA-Inf + ReVeil|0.60 / 0.00|0.76 / 0.04|0.70 / 0.04|0.64 / 0.02   |0.68 / 0.02|1.00 / 0.04|
>
>
> **In Table 7, the unlearning ratios are not relative to the whole training data, but they are relative to the clean data, which is a subset of the whole training data.** We followed the original unlearning attack settings and only set the maximal unlearning ratios to 50%. Specifically, for hidden poisoning attacks, the training dataset $\mathcal{D}$ consists of two parts: the poison dataset $\mathcal{D}\_{po}$ and the clean dataset $\mathcal{D}\_{cl}$. Therefore, the whole training data is $\mathcal{D}=\mathcal{D}\_{po} \cup \mathcal{D}\_{cl}$. Unlearning data $\mathcal{D}\_{f}$ is selected from $\mathcal{D}\_{mi}$​, *as illustrated in Lines 2937–2938 of Appendix in the initial submission.* Unlearning ratio is calculated as $\frac{|\mathcal{D}\_f|}{|\mathcal{D}\_{cl}|}$. Specifically, unlearning 50% of the clean data (520 samples) constitutes only 25% of the whole training dataset (1040 samples). **Additional experimental setup details are provided in Appendix Section 11.1 (the first and second paragraphs).** Finally, for your convenience, we also change results using the alternative absolute ratio $\frac{|\mathcal{D}_f|}{|\mathcal{D}|}$. As shown in Table 2, strong attack performance persists even under these small absolute unlearning ratios.

---

> ### Author Response · Authors · 2025-11-21
> **Response to Reviewer y9ov (Part 2/6)**
>
> Table 2: ASR (%) via the unlearning attack using different unlearning methods on different models under the revised unlearning ratio definition ($|\mathcal{D}\_f| / |\mathcal{D}|$).
> | Model | Unlearning ratio | GA | GA+GD | NPO | NPO+GD |
> | :--- | :--- | :--- | :--- | :--- | :--- |
> |Qwen2.5-1.5B|0%| $28.33 \pm 1.33$ | $28.33 \pm 1.33$ | $28.33 \pm 1.33$ | $28.33 \pm 1.33$ |
> | |5%| $13.00\pm5.51 $ | $27.00\pm11.06 $ | $40.00\pm1.73 $ | $51.00\pm6.66 $ |
> | |10%| $83.33\pm7.69$ | $79.33\pm5.36 $ | $46.33\pm3.48 $ | $64.33\pm5.93 $ |
> | |15%| $96.00\pm2.00 $ | $94.33\pm2.40 $ | $51.67\pm2.33$ | $76.67\pm0.67 $ |
> | |20%| $99.33\pm0.67$ | $97.33\pm0.88 $ | $52.00\pm1.53 $ | $79.33\pm3.93 $ |
> | |25%| $99.67\pm0.33$ | $97.67\pm1.20 $ | $51.33\pm2.33 $ | $87.67\pm0.33 $ |
> |Qwen2.5-3B| 0% | $0.67\pm0.33$ | $0.67\pm0.33$ | $0.67\pm0.33$ | $0.67\pm0.33$ |
> | |5%| $31.67\pm3.18$ | $8.67\pm4.41$ | $0.67\pm0.33$ | $0.67\pm0.33$ |
> | |10%| $87.67\pm1.33$ | $86.67\pm1.86$ | $0.67\pm0.33$ | $ 0.67\pm0.33$ |
> | |15%| $94.00\pm1.15 $ | $95.67\pm1.20 $ | $0.67\pm0.33$ | $3.33\pm1.45 $ |
> | |20%| $98.33\pm0.88 $ | $95.33\pm2.03 $ | $0.67\pm0.33$ | $8.00\pm1.53 $ |
> | |25%| $99.67\pm0.33$ | $97.33\pm0.88 $ | $2.33\pm0.88$ | $28.67\pm1.86$ |
> |Qwen2.5-7B| 0% | $8.67\pm1.33$ | $8.67\pm1.33$ | $8.67\pm1.33$ | $8.67\pm1.33$ |
> | |5%| $76.00\pm1.53 $ | $75.33\pm2.33 $ | $28.33\pm2.73 $ | $36.00\pm5.77 $ |
> | |10%| $93.33\pm4.18 $ | $89.00\pm4.04 $ | $62.00\pm2.65$ | $61.00\pm7.51 $ |
> | |15%| $96.67\pm2.33$ | $94.33\pm1.76 $ | $81.67\pm2.33$ | $49.00\pm7.64 $ |
> | |20%| $100.00\pm0.00$ | $98.00\pm1.00 $ | $87.33\pm1.76$ | $52.33\pm2.19 $ |
> | |25%| $100.00\pm0.00$ | $97.67\pm0.88 $ | $92.00\pm2.00 $ | $70.00\pm4.58 $ |
>
>
> **In Figure 7 of our initial submission, the unlearning ratios are not relative to the whole training data, but they are relative to the mitigation data, which is a subset of the whole training data** under hidden poisoning attacks. Specifically, for hidden poisoning attacks, the training dataset $\mathcal{D}$ consists of two parts, including poison dataset $\mathcal{D}\_{po}$ and mitigation dataset $\mathcal{D}\_{mi}$.
> Therefore, the whole training data is $\mathcal{D}=\mathcal{D}\_{po} \cup \mathcal{D}\_{mi}$. Unlearning data $\mathcal{D}\_{f}$ is selected from $ \mathcal{D}\_{mi}$​, *as illustrated in Lines 2937–2938 of Appendix in the initial submission.* For the selected unlearning data $\mathcal{D}\_{f} \subset \mathcal{D}\_{mi}$, the unlearning ratio is calculated as $\frac{\mathcal{D}\_{f}}{|\mathcal{D}\_{mi}|}$. As a result, even our "100%" unlearning ratio corresponded to removing only a small fraction of the entire training set; specifically, unlearning 50% of the mitigation data (520 samples) constitutes only 25% of the whole training dataset (1040 samples). **More experimental setup details have been provided in Appendix Section 11.1 (first and second paragraphs).** Lastly, for your convenience, we also report results using the alternative absolute definition $\frac{|\mathcal{D}\_{f}|}{|\mathcal{D}|}$. As shown in Table 3, strong attack performance can be achieved even under these small absolute unlearning ratios. For informative poisoning attacks, we followed the original unlearning attack settings in the prior work (Ma et al., 2024) to consider extreme attack settings, but we only set the maximal unlearning ratio to 50%. In practice, extreme unlearning attacks are still practically feasible, particularly when a data owner contributes a substantial portion of the training data. This broader perspective helps researchers and practitioners to develop a more complete understanding of the potential risks and vulnerabilities under extreme attack settings.
>
> Table 3: ASR (%) of training-unlearning attacks with poisoning on different unlearning methods under the revised unlearning ratio definition ($|\mathcal{D}\_f| / |\mathcal{D}|$).
> | Hidden poisoning | 5%   | 10%   | 15%   | 20%  | 25%  |
> | :---|:--- | :--- | :---| :--- | :--- |
> | GA     | 91.00 ± 3.61 | 95.33 ± 0.88 | 95.67 ± 0.88 | 96.33 ± 0.33 | 98.00 ± 1.15 |
> | GA+GD  | 83.33 ± 2.85 | 96.67 ± 0.88 | 94.33 ± 1.20 | 96.67 ± 1.20 | 97.33 ± 0.88 |
> | NPO    | 31.33 ± 2.91 | 29.67 ± 2.40 | 28.33 ± 3.18 | 30.00 ± 1.00 | 29.67 ± 0.88 |
> | NPO+GD | 35.33 ± 7.88 | 39.00 ± 4.36 | 42.33 ± 7.54 | 56.00 ± 2.52 | 77.33 ± 5.78 |
>
> Lastly, to enhance clarity, we have revised the unlearning ratios in the revised version so that they are now expressed relative to the entire training dataset (see Figure 7 and Figure 8 in the main manuscript, and Table 7 in Appendix).

---

> ### Author Response · Authors · 2025-11-21
> **Response to Reviewer y9ov (Part 3/6)**
>
> **Q2. Does the paper define and provide the problem formulation and threat models?**
>
> **A2.** Thank you very much for the insightful comment. First, we would like to clarify that **due to space limitations, we have deferred the defined problem formulations and threat models for each unlearning attack and defense to Appendix (please refer to *Section 9.2, Section 10.2, Section 11.2, Section 12.2, Section 13.2, Section 14.2, and Section 15.2*)**. Specifically, *Section 9.2 in Appendix* presents the defined formal problem formulation (*Sections 9.2.1, 9.2.2; Equations 11, 12, 15, 16, 18, 19, 20, 21; Algorithms 1, 2, 3*) and threat model (*second paragraph*) for the sole unlearning attack scenario; *Section 10.2 in Appendix* provides the corresponding defined formal problem formulation (*Sections 10.2.1, 10.2.2; Equations 23, 24, 27, 28; Algorithms 4, 5*) and threat model (*second paragraph*) for the unlearning–inference attack scenario; *Section 11.2 in Appendix* presents the defined formal problem formulation (*Equation 29; Example 11.1*) and threat model (*second paragraph*) for the training–unlearning attack scenario; and *Section 12.2 in Appendix* provides the defined formal problem formulation (*Equations 30, 31; Examples 12.1, 12.2*) and threat model (second paragraph) for the training–unlearning–inference attack scenario. Additionally, *Section 13.2 in Appendix* provides the defined formal problem formulation (*Equations 32, 33*) and threat model (*first paragraph*) for the grey-box attack setting; *Section 14.2 in Appendix* provides the defined formal problem formulation (*Equations 35, 36; Algorithm 6*) and threat model (*second paragraph*) for the black-box attack setting; and *Section 15.2 in Appendix* provides the defined formal problem formulation (*Equations 38, 39, 40, 41, 42, 43*) and threat model (*second paragraph*) for the evaluated defenses. These materials are organized according to our taxonomy of attack scenarios, which is defined by the stages involved (training, unlearning, and inference).
>
> **Q3. All possible experiment considerations (around 8400 potential experimental combinations for attacks) for the results: i) discuss the claim in the paper; ii) discuss the generalization behaviors for the precision and reliability of findings.**
>
> **A3.** Thank you very much for the valuable comments. First, we would like to first clarify that as claimed in the paper, **in our framework designs, we have already covered comprehensive and large-scale perspectives**: (i) Our framework formalizes multiple different unlearning attack paradigms, which span different stages (including the training stage, the inference stage, and the testing stage) of the machine learning pipelines. This goes significantly beyond evaluating only the unlearning stage and instead examines how unlearning interacts with other operational stages, offering a more comprehensive and large-scale view of potential vulnerabilities. (ii) Our framework has considered the attacker’s capabilities from three different settings: white-box, grey-box, and black-box. These diverse settings significantly broaden the coverage and generality of our evaluations. (iii) In addition to unlearning attacks, our framework also incorporates a comprehensive suite of unlearning defenses, covering both detection-oriented defenses and mitigation-oriented defenses. These perspectives ensure that the defense component of our framework is both comprehensive and large-scale, enabling systematic evaluation of defenses. (iv) We have adopted multiple heterogeneous LLMs with different scales and architectures in the evaluations. (v) We have included diverse datasets that differ in size, domain, task goals, and safety risks in the evaluations. (vi) We have considered multiple unlearning paradigms that perform LLM unlearning via different mechanisms in the evaluations. (vii) We have evaluated unlearning attacks and defenses via different evaluation metrics in the evaluations. These dimensions altogether demonstrate that our benchmark designs are comprehensive and intrinsically large-scale, as stated in the paper.

---

> ### Author Response · Authors · 2025-11-21
> **Response to Reviewer y9ov (Part 4/6)**
>
> Additionally, we would like to further clarify that **in the experimental setting combinations, to show transferability and generalization across different settings, our initial submission has already incorporated comprehensive and large-scale transferability studies (Figure 8, Figure 28, Figure 29, Figure 30, Figure 31, Figure 32, Figure 33, Figure 34, Figure 35, Figure 36, Figure 37, and Figure 39)** showing generalization performance across many different combinations. Further, for your convenience, we have moved the experimental transferability results in Figure 8 of the main paper into Table 4 below. Here, we illustrate the transferability of unlearning attacks across different model architectures and scales. As shown, malicious unlearning requests generated on a surrogate model remain highly effective when applied to other target models, consistently degrading their performance, even the architecture and scale of the source and target models differ. Additionally, in Table 5 below, we conducted experiments to verify the transferability of unlearning data (moved from Figure 39 of Appendix). Here, we utilized substitute models to generate adversarial unlearning data, which are non-training data for the target model. As shown, the effectiveness of transferred adversarial unlearning data also negatively impacts the target model’s performance. Furthermore, Table 6 and Table 7 show the transferability of unlearning attacks on additional datasets (moved from Figure 35 and Figure 36 of Appendix, respectively), showing the effective unlearning attack transferability using various datasets. Note that Figures 8, 28-37, and 39 from the initial submission now appear as Figures 9, 30-39, and 42 in the revised version.
>
>
> Table 4: Transferability (performance degradation ratio (%)) of unlearning attacks across different model architectures and scales.
> |Source model\Target model|Llama2-7B|Llama2-13B|Qwen2.5-1.5B| Qwen2.5-7B|Mistral-7B-v0.1|Mixtral-8x7B-v0.1|
> |:---|:---|:---|:---|:---|:---|:---|
> |Llama2-7B|100.00|99.53|100.00|92.31|100.00|100.00|
> |Llama2-13B|100.00|100.00|100.00|99.39|86.01|100.00|
> |Qwen2.5-1.5B|100.00|100.00|99.30|100.00|100.00|100.00|
> |Qwen2.5-7B|100.00|100.00|100.00|99.37|100.00|100.00|
> |Mistral-7B-v0.1|100.00|100.00|100.00|100.00|100.00|100.00|
> |Mixtral-8x7B-v0.1|100.00|100.00|94.76|100.00|100.00|97.42|
>
>
> Table 5: Transferability (performance degradation ratio (%)) of unlearning attacks across adversarial unlearning data.
> |Source model\Target model|Llama-2-7B-Chat|Llama-2-13B-Chat|Llama-3-8B-Instruct|Mistral-7B-Instruct-v0.1|Vicuna-7B-v1.5|Vicuna-13B-v1.5|
> |:---|:---|:---|:---|:---|:---|:---|
> |Llama-2-7B-Chat|91.67|46.67|80.95|56.25|100.00|91.67|
> |Llama-2-13B-Chat|91.67|46.67|61.90|6.25|75.00|58.33|
> |Llama-3-8B-Instruct|75.00|53.33|80.95|12.50|12.50|50.00|
> |Mistral-7B-Instruct-v0.1|83.33|26.67|100.00|100.00|87.50|91.67|
> |Vicuna-7B-v1.5|91.67|40.00|100.00|100.00|100.00|83.33|
> |Vicuna-13B-v1.5|91.67|40.00|85.71|50.00|100.00|91.67|
>
>
> Table 6: Transferability (performance degradation ratio (%)) of unlearning attacks on more datasets.
> |Dataset|Source model|Llama-2-7B|Llama-2-13B|Llama-2-70B|Qwen2.5-7B|Qwen2.5-14B|Qwen2.5-32B|Mistral-7B-v0.1|Mixtral-8x7B-v0.1|
> |:---|:---|:---|:---|:---|:---|:---|:---|:---|:---|
> |Tofu|Qwen2.5-14B|83.70|65.67|95.20|75.33|52.54|98.44|100.00|99.72|
> |MedQuad|Qwen2.5-14B|100.00|100.00|97.12|99.96|100.00|99.01|100.00|100.00|
>
>
> Table 7: Transferability (performance degradation ratio (%)) of unlearning attacks on more datasets.
> |Dataset|Source model|Llama-2-7B|Llama-2-13B|Llama-2-70B|Qwen2.5-7B|Qwen2.5-14B|Qwen2.5-32B|Mistral-7B-v0.1|Mixtral-8x7B-v0.1|
> |:---|:---|:---|:---|:---|:---|:---|:---|:---|:---|
> |Tofu|Qwen2.5-32B|66.70|87.13|94.36|96.95|47.10|60.62|100.00|100.00|
> |MedQuad|Qwen2.5-32B|99.96|100.00|97.12|98.45|100.00|89.38|98.20|100.00|
>
>
> Moreover, following your suggestions, we conducted new transferability experiments. We followed the same setting for exact unlearning attacks in Figure 2, and we considered new transferability experiments across different unlearning methods on MedQuad and the Llama2-7B, Llama2-13B, and Mistral-7B-v0.1 models. Specifically, we generated 5% malicious unlearning requests using GA and then transferred to other unlearning methods, including GA+GD, GA+KL, NPO, NPO+GD, NPO+KL, RMU, and WHP. The results are shown in Table 8. As shown, malicious unlearning requests generated using GA are also effective in degrading the model performance when applied with other unlearning methods. These transferability results have also been included in Figure 40a of Appendix Section 13.3.2.
>
> Table 8: Transferability (performance degradation ratio (%)) of unlearning attacks across different unlearning methods.
> |Model|GA+GD|GA+KL|NPO|NPO+GD|NPO+KL|RMU|WHP|
> |:---|:---|:---|:---|:---|:---|:---|:---|
> |Llama2-7B|28.90|35.50|24.31|22.04|17.57|13.52|25.15|
> |Llama2-13B|13.50|17.60|100.00|62.47|69.63|23.76|17.86|
> |Mistral-7B-v0.1|16.24|28.77|100.00|96.09|78.99|100.00|39.34|

---

> ### Author Response · Authors · 2025-11-21
> **Response to Reviewer y9ov (Part 5/6)**
>
> Further, following your suggestions, we conducted additional new transferability experiments. We followed the same experimental setting of targeted adversarial unlearning attacks (TAU) in Figure 4, and we considered new transferability experiments across datasets using SciQ and PIQA with the Llama-2-7B, Llama-2-13B, Qwen2.5-1.5B, and Qwen2.5-7B models. We generated 10% malicious unlearning requests on the SciQ and PIQA datasets using the GA unlearning method and then transferred them across each other. The corresponding results are shown in Table 9. From this table, we can see that malicious unlearning requests generated using a different dataset are also effective in degrading the model performance when applied to other datasets. These transferability results have also been included in Figure 40b Appendix Section 13.3.2.
>
> Table 9: Transferability (performance degradation ratio (%)) of unlearning attacks across different datasets.
> |Source dataset|Target dataset|Llama-2-7B|Llama-2-13B|Qwen2.5-1.5B|Qwen2.5-7B|
> |:---|:---|:---|:---|:---|:---|
> |SciQ|SciQ|46.25|45.82|39.54|34.72|
> |SciQ|PIQA|15.00|36.03|36.67|34.38|
> |PIQA|PIQA|19.96|29.04|23.33|25.00|
> |PIQA|SciQ|58.82|75.00|39.39|52.94|
>
>
> Lastly, we want to emphasize that these transferability analyses across various settings (e.g., various models, and various unlearning methods) collectively demonstrate generalization behaviors under heterogeneous conditions, thereby supporting the precision and reliability of our findings. Additionally, given the flexible and adaptive design of our benchmark, we will incorporate this helpful suggestion and continue to expand and refine it to accommodate broader evaluation settings.
>
> **Q4.  4.1) Moving results (on WHP and RMU) from Appendix to the main body. 4.2) Moving the experimental configurations from Appendix to the main text (unlearning procedures and the evaluation metrics) for more sufficient detail. Since they are not in the main text, the paper's presentation is poor.**
>
> **A4.** Thank you very much for the helpful suggestions. First, we would like to clarify that **due to space limitations, in the initial submission, we have deferred the experimental results on WHP and RMU and some experimental configurations to Appendix**. Following your suggestion, in the revised version, we have moved the WHP and RMU experimental results to the revised version (please refer to Figure 3 in the main paper for the experimental results, and lines 366–368 for the corresponding experimental settings and observations in the main text).
>
> Additionally, we would like to clarify that **in the initial submission, we have included some important experimental configurations, including the evaluation metrics in lines 346–349, in the main text**. Due to space limitations, we have deferred additional detailed descriptions to Appendix (*second paragraphs in Sections 9.1, 10.1, 11.1, 12.1, 13.1, 14.1, 15.1*). Following your suggestions and respecting the revision page requirements, we have moved some other details, including both the unlearning procedures and the evaluation metrics into the main manuscript (please refer to the third paragraph in Section 4.1, and the fifth and sixth paragraphs in Section 4.2 in the revised version).
>
>
> **Q5. Clarifying why the reported performance degradation in Figure 8 reaches 100%, calculated as $(P_f-P_u) / P_f$, where $P_f$ represents the original model performance and  $P_u$ the post-unlearning performance.**
>
> **A5.** Thank you very much for the precious comment. We would like to clarify that **in Figure 8 of our initial submission, we have considered the targeted attack success rate (ASR) over a set of attacker-selected victim samples from the target task** (instead of the full test data set). Our goal here is not to attack the model’s overall test accuracy across all test samples, but specifically to consider the attack effectiveness on these targeted victim samples. Specifically, for the performance degradation ratio $\left(P_f-P_u\right) / P_f$ shown in Figure 8, $P_f$ refers to the original model's performance on these victim samples on the target task, while $P_u$ denotes the performance on these victim samples after the adversarial unlearning update. **Note that we have also highlighted in blue that we in Figure 8 randomly selected 100 victim target samples (refer to Lines 3338 in Appendix).** According to Figure 8, the reported performance degradation reaches 100% and the model achieves 0% performance after unlearning, which indicates that our proposed unlearning attack is able to completely attack these target victim samples. Therefore, the results in Figure 8 are indeed intended outcomes to demonstrate the effectiveness of our proposed attack. Note that Figure 8 from the initial submission now appears as Figure 9 in the revised version.

---

> ### Author Response · Authors · 2025-11-21
> **Response to Reviewer y9ov (Part 6/6)**
>
> **Q6. Clarifying several comparison tables by indicating which ones are superior and highlighting key findings to guide readers on interpreting performance differences (e.g., higher vs. lower values).**
>
> **A6.** Thank you very much for the helpful suggestions. Following your suggestions, in the revised version, we have used clear indications to interpret the comparison tables (Tables 2, 3, 4, 5, 6, 7). Note that in Table 1 of Appendix, we provided a detailed taxonomy of the adopted unlearning attacks. Specifically, in the revised version, we have made the following revisions: (i) We have highlighted the best-performing results (marked in blue) and the worst results (marked in gray) in each of these comparison tables. (ii) For each comparison table, we have clarified the desired direction of each metric by using upward and downward arrows to indicate whether higher or lower values are better. (iii) We have explicitly highlighted the key findings by providing the relative ranking orders of the experiment configurations included in these comparison tables. Based on these revisions, we have also updated the table titles to better reflect the key comparisons and findings presented.
>
>
> **Q7. Improving the figure reference orders (Figure 5 and Figure 7).**
>
> **A7.** Thank you very much for the precious suggestion. First, we want to clarify that, given the extensive experimental results, **to make a deliberate trade-off between making figure references close to the corresponding experiment writing paragraphs for easier access and maintaining the overall figure reference orders**, we positioned these figures in our initial submission. For example, Figure 5 is placed on the right side of its related experiments in the second paragraph of Section 4.2, making it straightforward for readers to follow the experimental observations and verify the results. Additionally, following your suggestion, for Figure 5 and Figure 7, we have adjusted their figure references in the revised version. Note that Figure 5 from the initial submission now appears as Figure 4 in the revised version.
>
> **Q8. Describing the attack method (instead of only a reference) in Figure 5 and Figure 11.**
>
> **A8.** Thank you very much for the valuable suggestion. To clarify, in the initial submission, we in the main text have provided the attack method names corresponding to each reference for Figure 5 (Line 375-377) (non-targeted unlearning (NTU) (Hu et al., 2023), targeted unlearning (TAU) (Huang et al., 2024b), and partial unlearning (PU; Qian et al., 2023)) and Figure 11 (Line 505-506) (utility loss regularizer (ULR) (Oesterling et al., 2024) and the OBLIVIATE framework (OBL) (Xu et al., 2025)). Following your suggestion, we have updated Figures 5 and 11 by replacing the reference numbers with the corresponding method names. Note that Figure 5 and Figure 11 from the initial submission now appear as Figure 4 and Figure 12 in the revised version.
>
> **Q9. Improving the structure and flow of figures and tables in the paper for better readability and coherence.**
>
> **A9.** Thank you very much for the helpful suggestion. Following your suggestions, we have checked and revised the paper to improve the structure and flow of figures and tables. Specifically, we corrected the reference orders for Figures 19, 20, and 21  in Appendix of the initial submission. Note that Figures 19, 20, and 21 from the initial submission now appear as Figure 21, 23, and 22 in the revised version.
>
> **Sincerely yours,**
>
> **Authors of Paper 13906**

---

### Author Response · Authors · 2025-11-24
**Global Response to the Reviewers**

**Dear Reviewers,**

We would like to express our sincere gratitude to the reviewers for their valuable time and comments. We highly appreciate all the feedback and suggestions, which help improve our work. We are greatly encouraged that they recognized the significance of our studied problem of LLM unlearning security as an under-explored yet critical direction (Reviewers UUqT and dzcA), highlighted its uniquely impactful attack surface beyond traditional attacks (Reviewer dzcA), and underscored the value of our comprehensive attack and formalization and novel attack taxonomy (Reviewers dzcA and ddTi). We appreciate their recognition that our benchmark includes various unlearning defenses (Reviewers y9ov, UUqT, dzcA, and ddTi). We are grateful that they found our benchmark is well supported by comprehensive and large-scale experiments (Reviewers y9ov, UUqT, dzcA, and ddTi). We also appreciate positive comments on our presentation’s clarity and quality (Reviewers UUqT and dzcA), nice attack visualization (Reviewers dzcA and ddTi), and detailed appendix (Reviewers dzcA and ddTi). Lastly, we are thankful that they recognized LUSB as a valuable public contribution and a comprehensive benchmark that helps set the standard for future research (Reviewers UUqT and dzcA).

Additionally, we have provided point-by-point responses for reviewers. While we have provided the individual responses for the reviewers, we also wanted to summarize what has been pointed out in the reviews, for which we have offered corresponding responses.

* For **Reviewer y9ov**, the raised comments that we have addressed are given below.
  * Regarding providing more clarifications of unlearning ratios, problem formulation, and threat models, we have provided detailed respective explanations and clarifications in A1 and A2.
  * Regarding experiments on all possible experiment considerations, we have provided detailed responses in A3.
  * Regarding the suggestion on moving certain experiment results from Appendix to the main paper, we have followed your suggestion and also provided the individual response in A4.
  * Regarding providing clarifications on performance degradation in Figure 8, we have provided detailed clarifications in A5.
  * Regarding suggestions on the tables and figures, we have followed your suggestions and also provided detailed respective responses in A6, A7, A8, and A9.

* For **Reviewer UUqT**, the raised comments that we have addressed are given below.
  * Regarding providing more in-depth analysis of unlearning-reduced model robustness, our detailed responses can be found in A1.
  * Regarding the suggestion on proposing new unlearning methods, we have followed the suggestion and also provided detailed responses in A2.

* For **Reviewer dzcA**, the raised comments that we have addressed are given below.
  * Regarding providing more deep analysis of defense experiments, we have provided detailed clarifications in A1.
  * Regarding providing more details for practical threat attack models and more detailed discussions of the practical adversary for each scenario, our detailed respective responses can be found in A2 and A4.
  * Regarding further clarifications about "Training-Unlearning-Inference" attack, we have provided detailed respective responses in A3 and A5.
  * Regarding discussions on the query complexity of query-based attacks, we have provided detailed responses in A6.


* For **Reviewer ddTi**, the raised comments that we have addressed are given below.
  * Regarding providing discussions on fine-tuning attacks, we have provided detailed respective responses in A1, A9, and A10, where we have clarified that discussions on fine-tuning attacks and fine-tuning based experiments have already been included in our initial submission.
  * Regarding providing more clarifications on method formalisation and attack constraints, our detailed respective responses can be found in A2, A3, and A4.
  * Regarding suggestion on adopting attacks and defenses from the given reference, we have followed your suggestion and also provided detailed responses in A5.
  * Regarding suggestions on paper formatting, figure captions, and moving certain metrics and terms from Appendix to the main paper, we have followed the suggestions and also provided respective responses in A6 and A7.
  * Regarding clarifications on the coverage of all cited methods, our detailed response can be found in A8.
  * Regarding further clarifications on the motivation for establishing the unlearning attack benchmark, we have provided detailed responses in A11.


Further, based on reviewers’ comments, in the uploaded revised version, we have made corresponding changes and also accordingly highlighted them in blue for clarity.

We look forward to engaging in further discussions and answering any further questions that reviewers may have.

We are truly grateful for your valuable time and thoughtful consideration.

**Sincerely yours,**

**Authors of Paper 13906**

---

### Meta-Review · Area_Chair_Gkze · 2026-01-04

**Summary:**

This submission introduces LUSB (Language Unlearning Security Benchmark), aiming to formalize and systematically benchmark unlearning attacks and defenses for LLMs across multiple attack stages and safety threat types. The paper reports large-scale empirical evidence that unlearning attacks can significantly degrade model security, including amplifying jailbreak vulnerabilities and reactivating or strengthening backdoor behaviors, while also arguing that existing LLM security defenses are often ineffective under unlearning attacks, motivating the need for new unlearning-security–aware defenses.

Across reviews, there is high variance in perceived novelty and clarity. Two reviewers view the work as timely and impactful, emphasizing the importance of benchmarking unlearning-induced vulnerabilities and the breadth of the experimental sweep. Conversely, two reviewers raise major concerns that the contribution is primarily an empirical compilation with insufficiently clear/complete formalization, unclear positioning relative to harmful fine-tuning attacks, and presentation issues that hinder readability.

In the rebuttal, the authors address several interpretation concerns, including clarifying that some “100% unlearning ratio” settings are relative to an "attack mitigation subset" rather than the full dataset, and arguing feasibility for the query budget used in some attack settings; at least one positive reviewer explicitly states they maintain their score after the clarifications.

**Reviewer Concerns:**

**Reviewer dzcA**

Addressed: most questions were resolved in the rebuttal; reviewer explicitly states they maintain a positive score after the responses.

Outstanding: requests for additional analysis remain reasonable but not blocking given their stance.

**Reviewer UUqT**

Addressed: The rebuttal partially clarifies aspects of feasibility and framing, and the overall scope is viewed as important.

Outstanding: The reviewer would like deeper analysis of why unlearning reduces robustness and clearer guidance toward new unlearning strategies, beyond benchmarking.

**Reviewer y9ov**

Addressed: Authors clarify interpretational issues and attempt to improve exposition around evaluation setups.

Outstanding: Reviewer argues the paper is largely summarization without sufficient novelty, and raises concerns about clarity and whether the paper provides fundamentally new insights beyond assembling evaluations.

**Reviewer ddTi**

Addressed: Authors respond at length on positioning vs fine-tuning attacks and claim the paper already includes experiments connecting unlearning to fine-tuning attack settings.

Outstanding: Reviewer maintains that the formalization is insufficiently specified, the connection to the harmful fine-tuning literature is not properly integrated, and flags readability issues.

**Reviewer Scores:**

Reviewer dzcA: 8 (accept, good paper); Confidence 3; explicitly maintains positive score post-rebuttal.

Reviewer UUqT: 6 (weak accept); Confidence 2.

Reviewer y9ov: 2 (reject, not good enough); Confidence 4.

Reviewer ddTi: 0 (strong reject); Confidence 3.

---

### Decision · Program_Chairs · 2026-01-26

Reject